# Accumulation and Cross-Shelf Transport of Coastal Waters by Submesoscale Cyclones in the Black Sea

Arseny Kubryakov *[ID], Anna Aleskerova, Evgeniy Plotnikov, Artem Mizyuk, Alesya Medvedeva and Sergey Stanichny

Marine Hydrophysical Institute, Russian Academy of Science, 2 Kapitanskaya ul., 299011 Sevastopol, Russia; annete08@mail.ru (A.A.); ev.plotnikov@yandex.ru (E.P.); artem.mizyuk@yandex.ru (A.M.); suomi-npp@mail.ru (A.M.); sstanichny@mail.ru (S.S.)

* Correspondence: arskubr@mhi-ras.ru

**Abstract:** High- and medium-resolution satellite optical imagery show that submesoscale cyclonic eddies (SCEs) trap coastal waters and induce their rapid cross-shelf transport. Due to the presence of a rigid boundary, the convergence is observed in the coastal part of SCEs. It causes accumulation of suspended matter, which spins inward in a spiral motion toward the SCE core. Small SCEs with a radius of 1–10 km transport waters with local anomalies in the concentration of chlorophyll, total suspended matter and temperature to a distance of up to 150 km and are observed for more than 10 days. Lagrangian calculations based on realistic NEMO numerical model are used to estimate the fate of the coastal waters in such SCEs. The eddy entrains the largest number of particles during its separation from the coast when its vorticity reaches the maximum. Then, the SCE weakens, which is accompanied by the flattening of initially risen isopycnals and deepening of the trapped coastal waters. The described mechanism shows that coastal SCEs may cause intense short-period cross-shelf transport of the biological and chemical characteristics, and is another process affecting the functioning of the marine ecosystems.

**Keywords:** submesoscale eddies; cross-shelf transport; total suspended matter; convergence; eddy transport; coastal zone; Black Sea





## 1. Introduction

Convergent/divergent motions in eddies, accompanied by intense vertical motions, significantly impact the transport of matter in oceans. Closed streamlines in the mesoscale eddies allow them to effectively trap the fluid and dampen the mixing between the eddy interior and the surrounding ocean [1,2]. Particularly, mesoscale anticyclones can trap and transport waters to more than 1000 km away from its origin, creating strong local anomalies in the biological, chemical and physical properties [3].

Submesoscale eddies (SEs) are one order smaller O (1–10 km) than the mesoscale, having approximately the same orbital velocity (~0.1 m/s) [4]. Thus, they are often characterized by very high values of vorticity [5], which in some extreme cases can exceed 10f [6], where f is Coriolis frequency. Such a strong vorticity probably intensifies the transport barrier between the eddy core and the surrounding waters, making SCEs even more effective for the accumulation and transport of fluid in oceans [7].

One important difference in the dynamics of meso- and submesoscale eddies is related to the larger contribution of centrifugal forces to SE dynamics. This effect leads to rather interesting changes in eddies' properties. One such feature is the dominance of submesoscale cyclonic eddies (SCEs) in satellite images [8–10], which is also confirmed with the results of numerical models [11]. Strong submesoscale anticyclones (SAEs) are also detected [6], but much less frequently than SCEs, which can be explained by the impact of centrifugal forces pulling the waters out of SAEs, thereby making them unstable and

short-lived [4,12]. In contrast, we often observe a larger number of stronger AEs compare to CEs on mesoscales [13,14].

Submesoscale dynamics are actively investigated in the Black Sea using in-situ [4,15], satellite [9,10,16–21], and numerical modeling data [22–24]. These investigations demonstrate that the orbital velocity of small SEs with a radius of 1–5 km may reach 0.1–0.2 m/s, while their Rossby number is about 1f [4,10], which can sometimes reach more than 10f according to drone measurements [6]. SCEs are most abundant in the coastal areas (Figure 1), where their generation is often related to the interaction of mesoscale eddies or alongshore currents with capes [4,10], instabilities on the front of coastal upwellings and river plumes [10,25]. This is distinctly observed on the probability map of SCE detection (P) obtained on the basis of high-resolution modeling in [10] (Figure 1b). Generally, P in the coastal area is 5–10 times higher than in the open sea. The hot-spots of SCE generation are concentrated near the capes, where P on average is 0.1–0.2, i.e., SCEs are detected 35–70 days per year. In comparison, for the open sea, these numbers are threefold smaller.

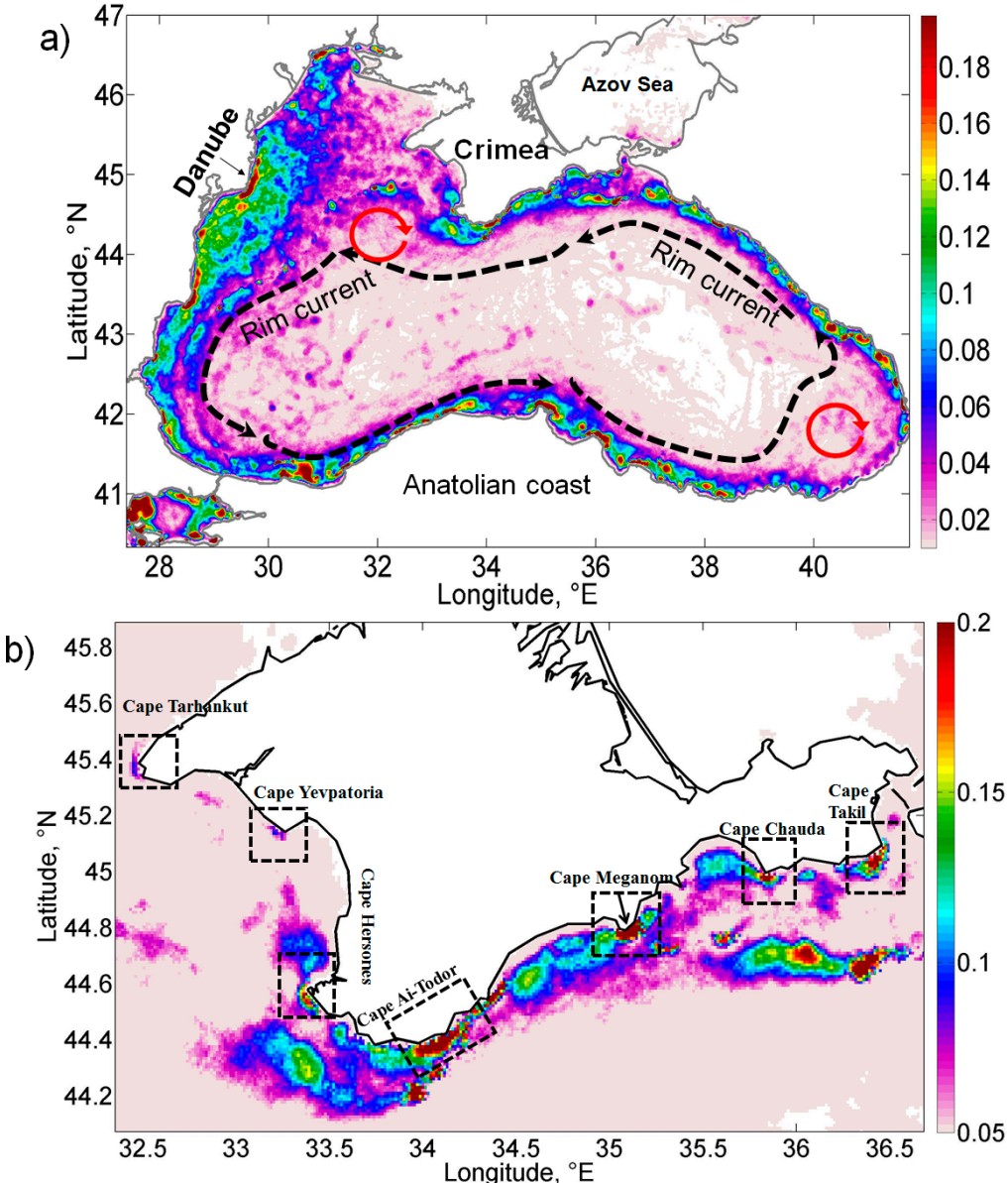

**Figure 1.** The probability of the identification of eddy with radius < 10 km on the basis of NEMO numerical model data in the Black Sea (**a**) and zoomed on Crimean coast from [10] (**b**).

However, only a few studies have investigated the impact of submesoscale eddies on transport in oceans. Particularly, [7] demonstrates that submesoscale eddies are an important element transporting deep waters from the convection region in the Mediterranean Sea. Authors of [26] investigated the impact of submesoscale eddies on the transport of anomalously cold water in the Sea of Japan. In [27,28], the authors demonstrated the impact of SCE on the transport of ice and heat in high latitudes. Several recent studies show that coastal submesoscale eddies in the Black Sea can cause strong local anomalies in the optical properties of the basin related to the entrainment of coastal waters in their core [10,15,26,29]. The offshore transport of shelf waters is one of the important sources of nutrients for the biota of the Black Sea deep waters, and can significantly affect its salt balance [30–32]. At the same time, investigations of the impact of small submesoscale eddies (with size 1–10 km) on the lateral transport of shelf waters in their core is almost absent.

In this study, we use high- and medium-resolution satellite measurements to investigate the impact of small SCEs on cross-shelf transport of coastal waters in the Black Sea. Satellite data show that in contrast to what is observed on the mesoscales, SCEs cause convergent spiral motions, which lead to the accumulation of coastal waters in their core (see Sections 2 and 3). Such an effect is caused by the impact of coast, which plays the role of a wall, and cause the convergence near the coast. Further, such SCEs can move rapidly to the deep sea via the background currents to a distance of 100–200 km. Here, they cause strong local anomalies in the concentration of total suspended matter (TSM) or chlorophyll a (Chl), which gradually decrease with time (Section 4). The cross-shelf transport of SCEs is reproduced in realistic numerical modeling (Section 5), which allows to estimate the amount and the fate of the coastal waters entrained in such SCEs.

## 2. Materials and Methods

High-resolution Sentinel-2 and Landsat imagery is used to study the small-scale structure of submesoscale eddies. Data were downloaded from https://www.sentinel-hub.com/ (accessed on 23 July 2023). To estimate the dynamic characteristics of SCEs from satellite data, we used an optical flow method called 4-D variational assimilation algorithm [33]. The algorithm is based on the analysis of the normalized radiance of two quasi-synchronous images of Landsat-8 and Sentinel-2 obtained within a time interval of several minutes. The resolution of the reconstructed velocity fields is 100 m. The details of the algorithm and data processing can be found in [10,33,34].

Modis-Aqua Level 2 daily data on remote sensing reflectance and chlorophyll A with 1 km resolution were used to describe the impact of submesoscale eddies on the optical properties. Data were downloaded from OceanColor data archive (http://oceancolor.gsfc.nasa.gov/) (accessed on 23 July 2023). Total suspended matter (TSM) was calculated from MODIS reflectance measurements using the regional Black Sea algorithm described in [35]. The algorithm is based on MODIS reflectance measurements at 443, 488 and 547 nm. It was developed on the basis of comparison with in situ measurements of TSM obtained in several surveys of the P. P. Shirshov Institute of Oceanology of the Russian Academy of Science. In situ data for validation of this algorithm were obtained in both clean waters of the open sea and very turbid coastal waters near the river plumes [36].

Geostrophic velocities were computed based on the regional Black Sea array of mapped altimetry sea level anomalies produced by Ssalto/Duacs and distributed by Copernicus Marine Service. Regional array has an improved spatial resolution of 1/8° and daily temporal resolution.

We also use the results of numerical modeling based on the NEMO modeling framework. The regional configuration was developed to reproduce the meso- and submesoscale variability in the Euxinus cascade basin, which includes the Azov, Black, and Marmara seas [37,38]. This model is able to reproduce the main features of the Black Sea circulation and the vertical thermohaline structure [38]. The verification of the model's results with satellite optical data shows that this model allows us to reproduce the main features of the dynamics of the mesoscale and submesoscale eddies. The horizontal grid is a quasi-uniform

geographical mesh with a resolution of $1/96° \times 1/69°$ northward and eastward, respectively. The model bottom topography is based on bathymetric data from the EMODnet v1 digital elevation model (URL: http://www.emodnet-bathymetry.eu, accessed on 23 July 2023). The calculation was carried out for the period 2008–2009. For vertical discretization, we used a partial step z-coordinate. A time step of 1 min was chosen. Surface boundary conditions were obtained via the product of ECMWF ERA5 reanalysis with a spatial resolution of $1/4°$ and a time resolution of 1 h. Vertical turbulent mixing was performed using the k–ε model [39]. The lateral diffusion of momentum and tracers were described with a bilaplacian operator. The values for turbulent viscosity and diffusivity are $(-4 \cdot 10^7 \, m^4/s)$ and $(-8 \cdot 10^6 \, m^4/s)$ [37], respectively. Advection terms of tracer equations are calculated with the TVD scheme [40]. The UNESCO formula is used as the equation of state. A more detailed description of the experiment is presented in [37]. The results of the numerical calculation for the study area are freely available at (https://zenodo.org/record/5607591, accessed on 23 July 2023).

## 3. Results

### 3.1. Accumulation of Suspended Matter in Submesoscale Cyclones

One of the known reasons for the intense formation of SCEs is related to the high horizontal shear and gradients of vorticity on the frontal zone of the mesoscale anticyclones [4,10,16,41,42]. Figure 2 presents an example of such a process observed on high-resolution Landsat image for 9 April 2019. A near-shore mesoscale anticyclonic eddy (MAE) with a radius of 45 km is situated near the south coast of the Black Sea (Figure 2a). MAE traps the suspended matter from the coast in its orbital motions on the western periphery and transports it to offshore. From the corresponding VIIRS map of total suspended matter (TSM), it is seen that TSM near the coast at this time is more than 2 mg/L (Figure 3). During its advection by MAE, TSM decreases to 0.7–1 mg/L on the western periphery of MAE, while the background concentration in the deep sea is about 0.4 mg/L. The interaction of MAE with the coast causes the formation of a series of SCEs. At least five SCEs are observed on the periphery of MAE in Figure 1.

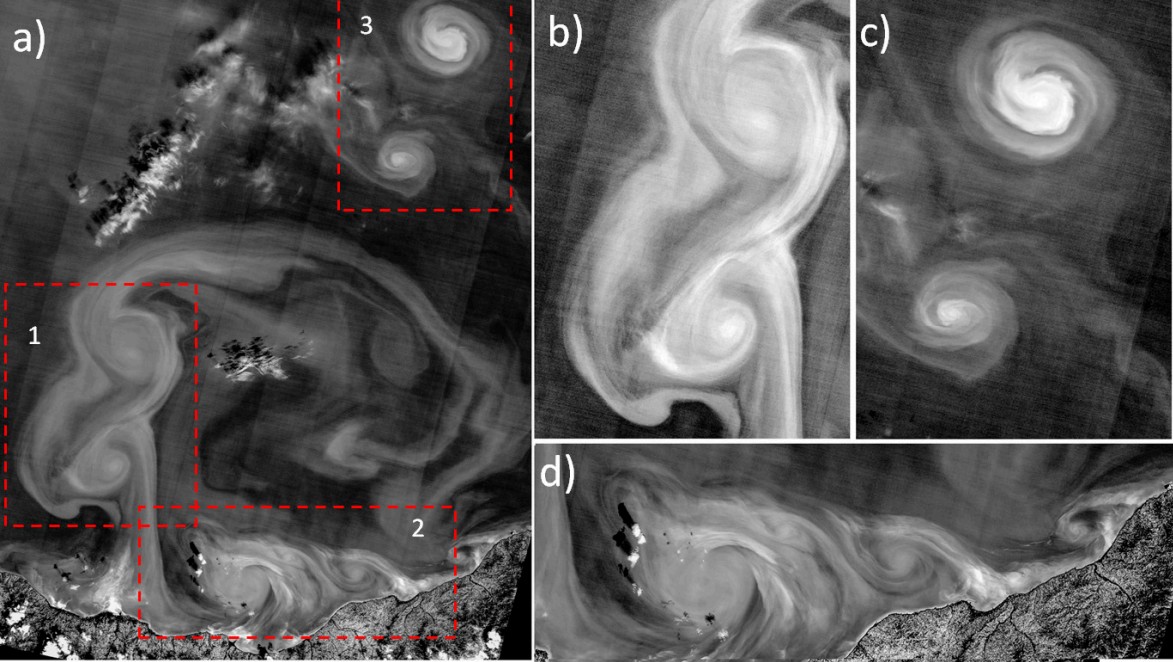

**Figure 2.** (**a**) Landsat-8 map of reflectance in panchromatic channel for 9 April 2019 in the southeastern part of the Black Sea; (**b–d**) zoomed images for the areas shown in Figure 1a by red rectangles. (area 1 is shown in Figure 1b; area 2—in Figure 1d, area 3 in Figure 1c).

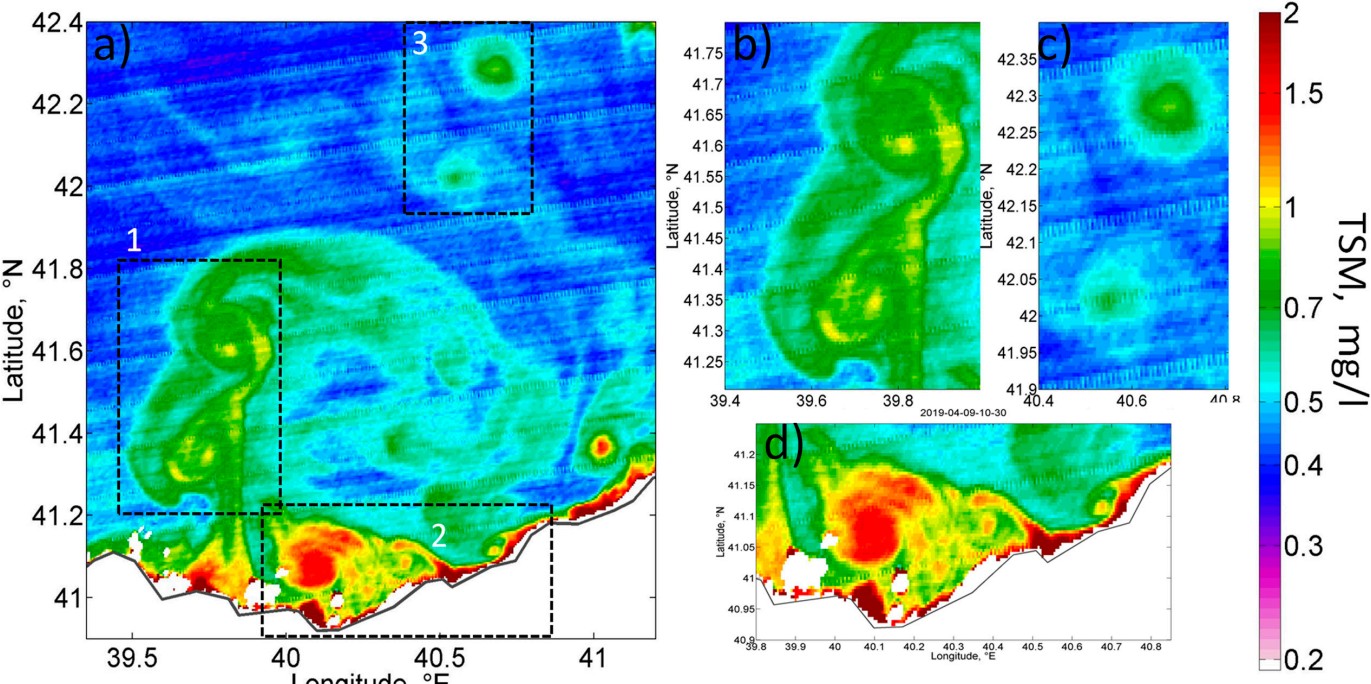

**Figure 3.** (**a**) Viirs map of total suspended matter (mg/L) for 9 April 2019 in the southeastern part of the Black Sea; (**b–d**) zoomed images for the areas shown in Figure 2a by red rectangles. (area 1 is shown in Figure 2b; area 2—in Figure 2d, area 3 in Figure 2c).

Two largest SCEs (Figure 2b) are situated in the western part of MAE. They have a radius of 10 km, two times less than the baroclinic radius of deformation in the Black Sea (Rd = 20 km). Figure 3b demonstrates the details of TSM distribution in these eddies. From the Landsat and Viirs images, it is seen that SCEs trap a part of TSM transported by anticyclone. Turbid waters are transferred along a spiral trajectory to the SCE core with a size of 1–5 km. Viirs data provide quantitative estimates of TSM in the core of this SCE, which reaches ~1 mg/L, corresponding to the maximum TSM at the MAE western periphery (Figure 3b).

Three SCEs are observed in the south coastal part of MAE (Figures 2d and 3d). The size of these eddies increase as they are transported to the west by MAE: smallest eddy with a radius of 2 km is situated in the eastern coastal part, second eddy has a radius of 7 km and the third is a westward SCE with a radius of 10 km. It is seen that these SCEs also entrain turbid coastal waters in cyclonic orbital motions. Then, these turbid waters make a spiral inward cyclonic turn and finally accumulate into their cores.

To the northeast of the MAE, we can observe two other remarkable SCEs (Figures 2c and 3c). Analysis of Viirs data shows that these eddies were also previously attached to the same MAE. Then, they separate from it and move to the open sea. Their radii are about 10 km, which is comparable to the size of eddies in Figure 1b. Both eddies are characterized by high reflectance, that is caused by the higher amount of TSM in SCE compared to the surrounding waters (Figure 3d). In Viirs data, these SCEs are seen as spots with increased TSM (0.5 mg/L), reaching maximum (1 mg/L) in the eddy cores. Landsat data show that the cyclonic orbital motions in these SCEs transport the most turbid waters to the eddy center. The presented example evidences that SCEs can trap and retain suspended matter in their cores.

### 3.2. Convergence in the Coastal Part of Submesoscale Cyclones

Consecutive high-resolution images of Sentinel-2 and Landsat-8 with a time interval of about 15 min were used to reconstruct the dynamic structure of coastal SCEs. Three situations chosen for the analysis are shown in Figure 4. All of them represent coastal SCEs,

which entrain turbid coastal waters in their orbital motion. It is seen that turbid waters in all of these eddies move along the cyclonic spiral to the center of the eddy. Such an inward spiral movement indicates a convergence in SCEs.

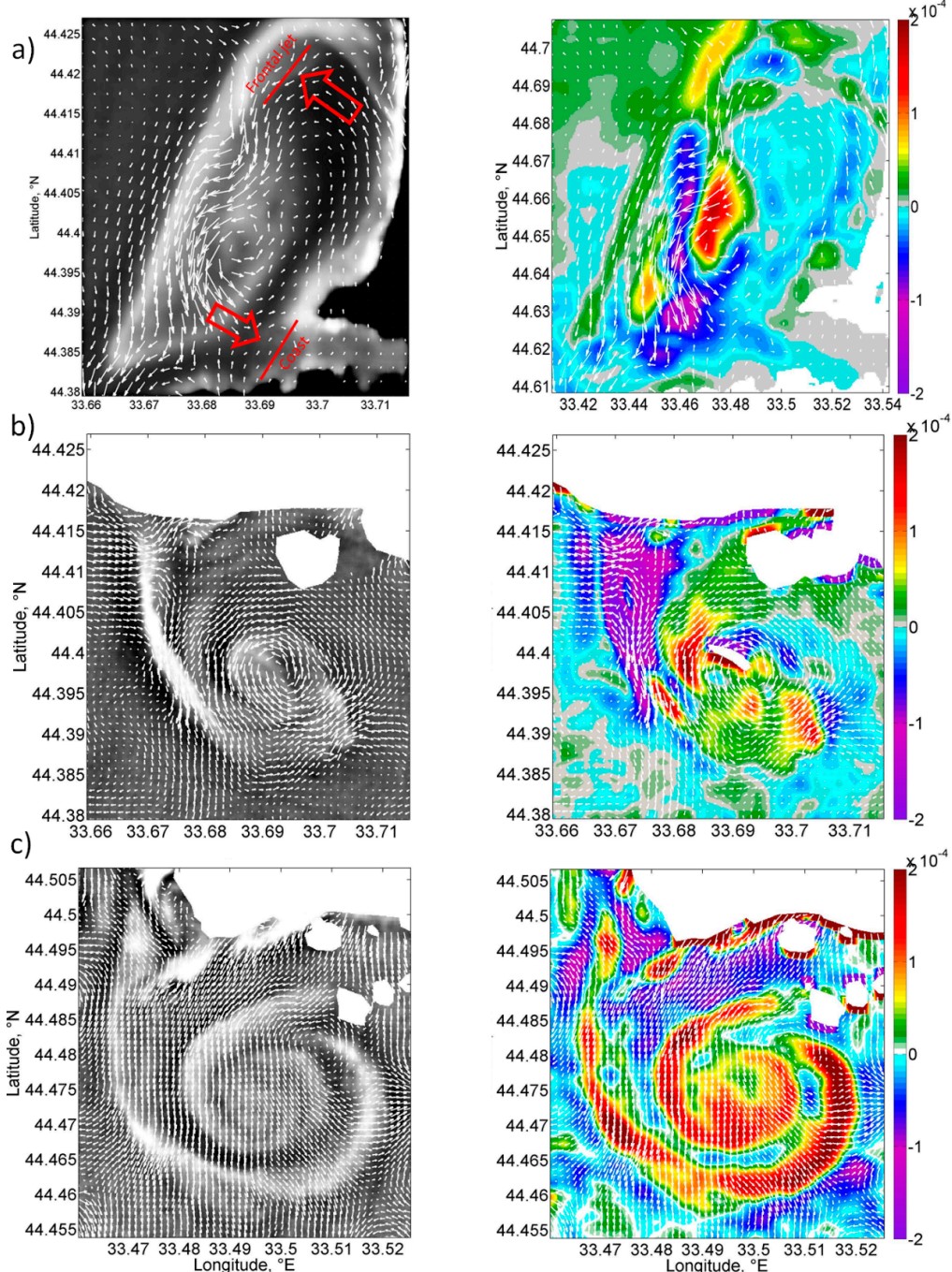

**Figure 4.** Left–normalized reflectance and right–computed divergence (1/s) for submesoscale cyclonic eddies observed on (**a**) SCE1—3 August 2020; (**b**) SCE2—26 February 2017; (**c**) SCE3—26 February 2017 near the Crimean coast. Arrows show the computed vectors of the current velocity. Red arrows in Figure 2a show the areas of convergence caused by the interaction of orbital currents with "walls" coast and frontal jet.

The reason for such convergence can be easily understood by looking at the example in Figure 4a. The turbid jet goes offshore from the coast, spins cyclonically and turns back to the coast. Here, the coast blocks the divergent motions in CE, and the turbid jet have to

turn back in the eddy center. In this area, the motion of the turbid waters is blocked by the southward currents in the outward spiral arm of the CE, which transport these waters to the south. As a result, an inward spiral pattern is formed. Sometimes, as in an example in Figure 4c, the jet moves inward along a spiral and then is blocked again by the previous spiral arm. So, this process repeats and the jet continue to move to the eddy center. Similar spirals are seen on all other examples, and for the coastal eddies, they are seen in Figure 2.

Therefore, the main reason of the observed convergence in SCEs is the presence of a rigid boundary on one side and frontal jet on the other side. The accumulation of suspended matter in the eddy is a result of fluid continuity: the interaction of the onshore currents with the "wall" causes downwelling and convergence in this part of the coastal SCE.

Distribution of velocities computed using 4-D variational assimilation algorithm is shown by arrows in Figure 3. The orbital velocity at the periphery of these eddies reaches about 0.2–0.4 m/s. The radius of these SCEs is ~5 km, respectively, that corresponds to the value of Rossby number ~1, indicating strongly nonlinear dynamics of these SCEs.

The data on surface velocity were used to compute divergence (*div*) in SCEs (Figure 4 right). The results confirm that divergence patterns in the eddy have a complex structure with different signs in the central and coastal part of the eddy. Particularly, for example 1 (Figure 4a), the divergence change in the area where the currents are directed toward onshore. The interaction of onshore currents with the coast may cause downwelling, convergence (*div* < 0) and accumulation of TSM, which is entrained in the eddy orbital motions from the north. The convergence is also observed in the inner zone of the CE, where currents are directed to the north and interact with the outward frontal jet. At the offshore periphery of SCEs, we observed divergence which is probably related to the strong impact of centrifugal motion and Coriolis force in SCEs.

Somewhat similar distribution is observed in case 2 (Figure 4b). The negative values (convergence) are detected in the part of the SCEs where currents are directed onshore, and positive values (divergence) are detected on the offshore periphery of the eddy. Another area of a strong convergence is observed in the area where coastal waters are entrained in the eddy orbital motion. Such eddies are mostly formed due to the interaction of eastward along-shore currents with capes (see more details in [10]). At the same time, cyclonic eddies induce the westward along-shore currents. The confluence of these currents is observed in the area of the current separation from the cape, where we observe the strongest convergence in Figure 4b,c. This effect promotes the entrainment of TSM in the eddy and can explain the maximal value of reflectance on the Landsat images (Figure 4b-left).

Third example demonstrates a more complex structure of divergence, which is probably related to several spiral arms observed in this eddy. The convergence is detected in the coastal part of this SCE and several patterns of divergence are observed in its offshore part. Also, weak divergence is observed between spiral arms in the core of the SCEs. This effect is often observed in SAR data, where eddies look like a spiral with repeating bands of convergence and divergence [8,9,43,44].

Not only the coast may act as a wall for the cyclonic spiral motions, but also similar features are observed on the front of intense, strong jet current. Figure 5 shows an example of such a situation observed on the Landsat-8 image on 30 August 2012. Highly turbid jet with a width of 10 km is seen near the western coast of Crimea. This jet was formed due to the action of the strong northeast winds, which cause upwelling and intense along-shore currents, transporting turbid coastal waters to the south [45]. Rapid change in the coastline orientation causes the separation of the jet from the coast, accompanied by its instability and the formation of SCEs [10,46]. Such SCEs trap a part of the turbid waters. These waters spin cyclonically and collide with the southward jet current. The jet blocks the westward motion in SCEs and causes the southward advection of turbid waters back to the SCE core. As a result, the inward spiral forms, which cause the propagation of turbid water in the center of such SCEs (see red rectangles in Figure 5b).

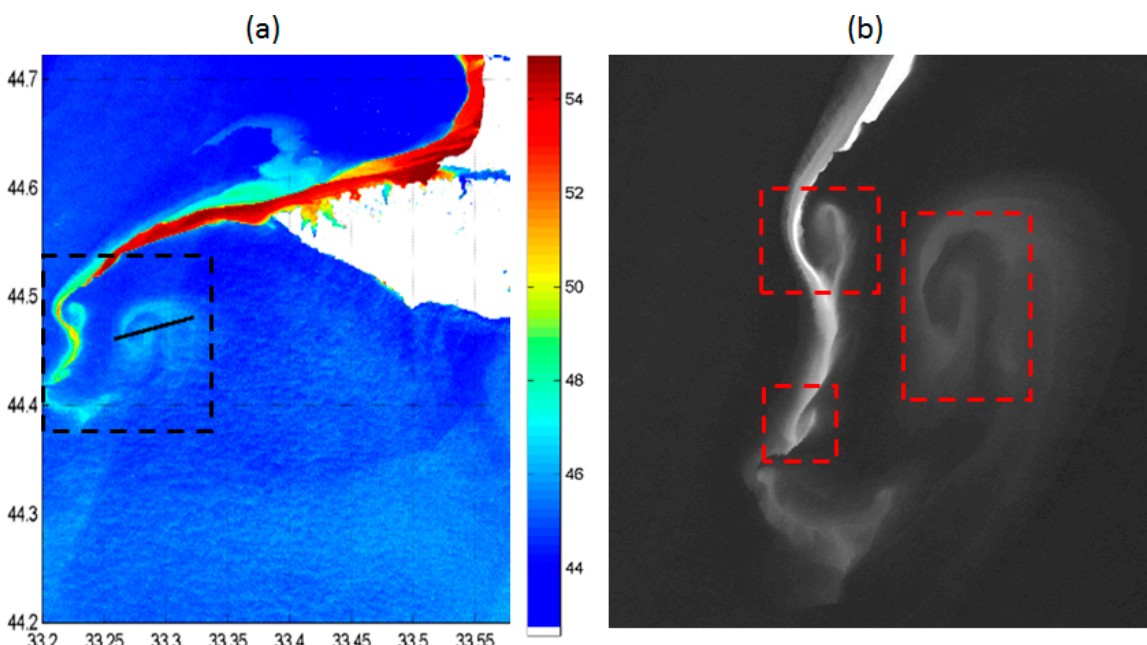

**Figure 5.** The inward spiral motion of turbid waters in submesoscale cyclonic eddies formed due to the instability of intense frontal current: (**a**) radiance (Watts/m$^2$/srad/μm) from Landsat-8 image for 30 August 2012; (**b**) zoomed image of the frontal jet, showing 3 SCEs (red rectangles).

The conceptual model of the discussed process is shown in Figure 6 [15]. SCEs are characterized by intense radial velocities directed out of their center, which are related to the action of Coriolis and centrifugal forces. Therefore, in the offshore part of the SCEs, divergence is observed, which is in agreement with the analysis of high-resolution data (letter D). However, near the coast, these motions are blocked by the "wall". Due to fluid continuity, this process leads to convergence (letter B) and downwelling near the coast (letter A). SCEs near the Crimean coast are often formed due to interaction of the along-shore jet (letter C) and capes [10]. Another area of convergence is formed in the place where the returning current in SCEs interacts with the along-shore jet. Such an effect is probably observed in Figure 3b. As a result, the along-shore jet spins around SCEs and entrains in its center between the areas of convergence and divergence (letter E). This process is the probable reason for the observed accumulation of suspended matter in the SCE core.

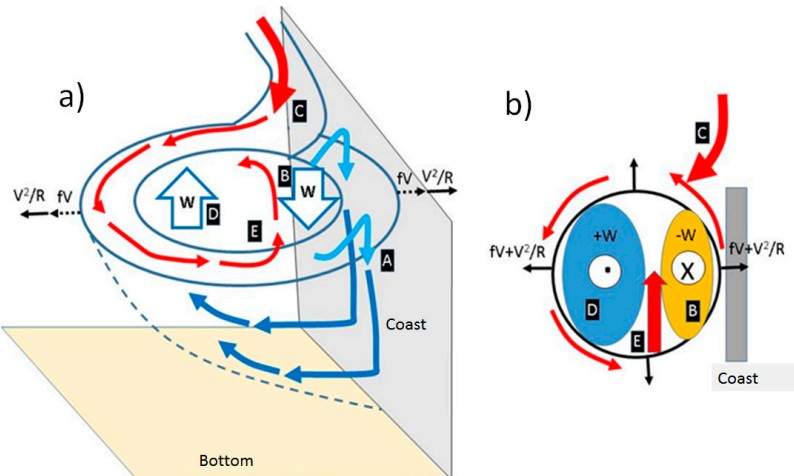

**Figure 6.** Schematic presentation of the convergence in the onshore part of SCE caused by the interaction of its orbital motions with the coast: (**a**) side view; (**b**) view from above (W—vertical velocity).

It should be noted that the structure of the divergence of SCEs is also affected by the presence of other factors, e.g., shear of background currents and the stage of evolution of these eddies (intensifying or decaying eddy), and thus, the structures can be very complex.

### 3.3. Transport by Submesoscale Eddies from Satellite Data

#### 3.3.1. Case 1. August 2020

The example in Figure 7a shows the series of SCEs formed on the periphery of MAE near the west Crimean coast on the high-resolution Sentinel image on 4 August 2008. Two SCEs detach from the coast and move in the westward direction. At this time, the third eddy, shown in detail in Figure 3a, is situated near the coast and accumulates coastal suspended matter.

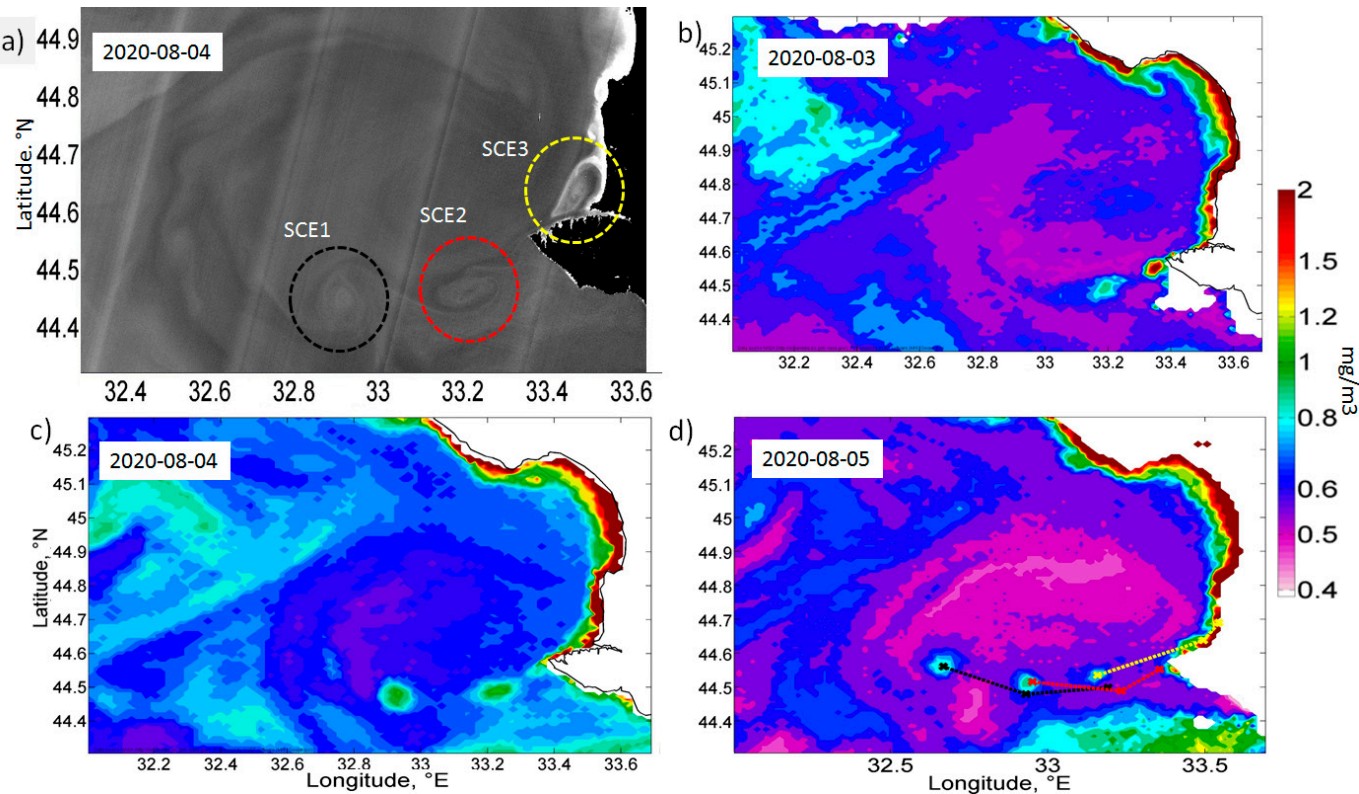

**Figure 7.** Evolution of series of SCEs attached to mesoscale eddies on (**a**) Sentinel-2 image for 4 August 2020; Modis maps (**b–d**) of concentration of chlorophyll A for (**b**) 3 August 2020; (**c**) 4 August 2020; (**d**) 5 August 2020. Rounds in Figure 6a show the position of the SCEs. Lines in Figure 6d show their trajectories.

In medium-resolution MODIS data, these eddies (Figure 7b–d) look like a round spot with increased concentration of chlorophyll A (Chl). Consecutive daily MODIS images can be used to track the evolution of such SCEs. On 3 August (Figure 7b), we observe the first SCE with Chl = 1 mg/m$^3$ separated from the coast. The second SCE is situated near the cape with twofold higher values of Chl = 2 mg/m$^3$. This value corresponds to the Chl near the coast and indicates that this SCE traps the productive coastal waters. On 4 August (Figure 7c), corresponding to Sentinel-2 image (Figure 7a), these SCEs displace westward. The first eddy move 30 km with a translational velocity of 25 cm/s and reaches 32.9°E. This velocity is close to the estimates of orbital velocity of the parent MAE from satellite altimetry data. The second SCE separates from the coast. Chl in this eddy decreases and reaches close to the value of Chl in the first eddy (1 mg/m$^3$). The radius of the core of these eddies at this time was about 5 km. On 5 August (Figure 7d), the third SCE separates from the coast, while the first SCE displaces further to 32.65°E. The scale of the local spots of the

Chl anomaly in this SCE decreased from 5 km to 3 km. On 6 August (not shown), all SCEs displace further to the west and the first eddy reaches 32.4°E. Then, the eddies dissipate.

In total, these SCEs exist for more than 4 days and move toward the distance exceeding 100 km from the coast. These examples demonstrate that the SCEs formed on the periphery of MAE induce transport of productive coastal waters in the deep part of the basin. The average translational velocity of these SCEs was about 0.25 m/s, which was close to the estimates of orbital velocity of the parent MAE from satellite altimetry data.

### 3.3.2. Case 2. September 2004

Another example of such a process was observed in MODIS data in September 2004 (Figure 8). On 10 September 2004, strong northeast winds with a velocity of 8–12 m/s hit the west coast of Crimea. Wave-driven erosion of the clay cliffs causes a release of a high amount of suspended matter off west Crimea [45]. TSM in the coastal zone after this storm in several places reached 10 mg/L, and was higher than 1 mg/L. At the same time, the values of TSM in the deeper part of the basin did not exceed 0.3 mg/L.

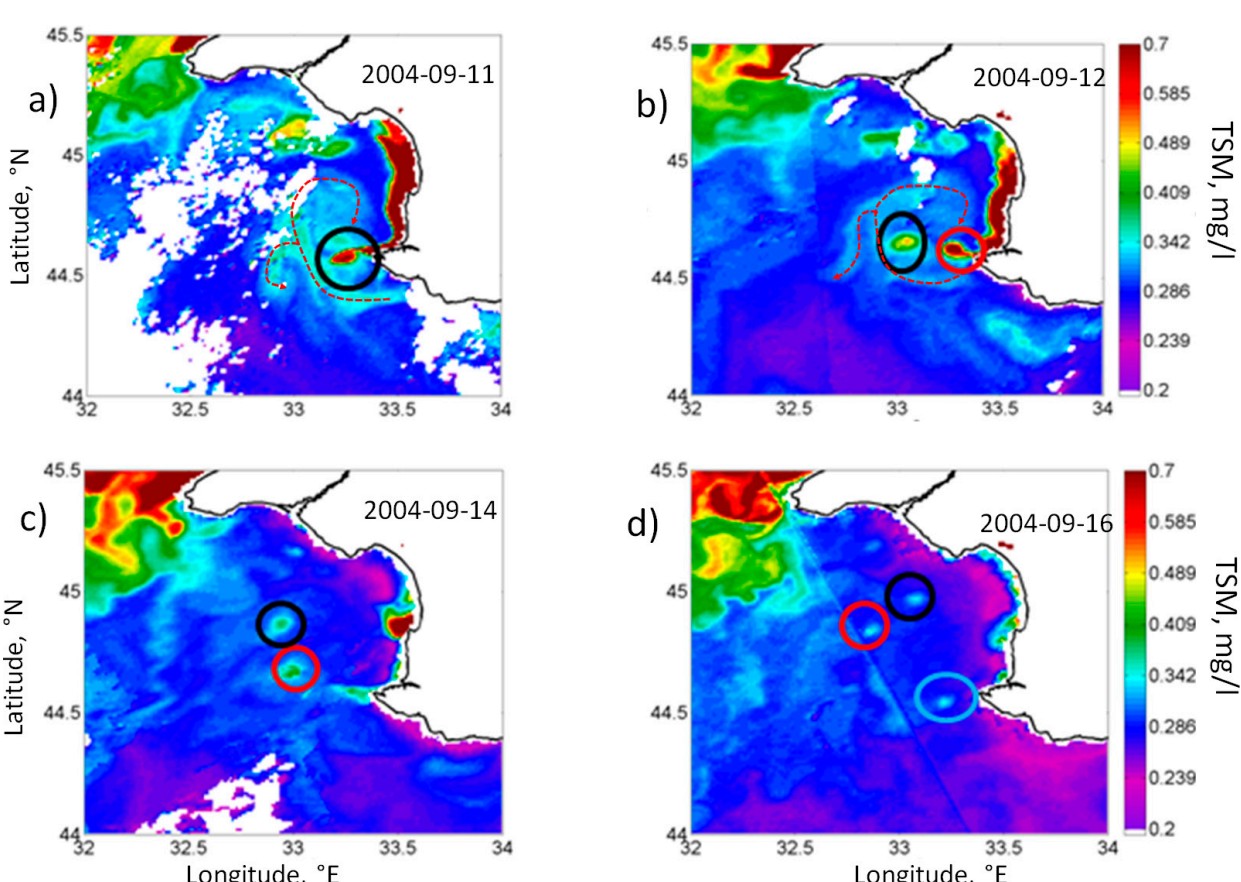

**Figure 8.** MODIS maps of TSM (mg/L) for (**a**) 11; (**b**) 12; (**c**) 14; (**d**) 16 September 2004 showing the generation of a series of submesoscale SCEs, their separation from the coast and transport of TSM in the deep part of the basin. Red arrows show the position of the mesoscale vortex dipole.

Strong northeast wind leads to the formation of southward coastal jet along west Crimea [45]. On 11 September 2004, the jet separates from the coast rotated cyclonically and formed the first SCE (C1) near Cape Hersones. In MODIS image, the eddy looks like a spot with increased TSM and a radius of 5 km. TSM in its core was high (>0.7 mg/m$^3$) and corresponded to TSM values in the along-shore jet near the coast.

On the next day, 12 September, this SCE separates from the coastal jet and displaces to the northwest at a distance of 25 km. TSM in its core decreased from 0.75 mg/L to 0.5 mg/L (Figure 8b). At the same time, a second eddy (C2) was formed near Cape Hersones. TSM

values in this SCE were high (>0.7 mg/m³), which correspond to TSMs near the coast. On 16 September, a third SCE (C3) of similar size is formed in the same area. The value of TSM in its core was somewhat less than in C1 and C2 (~0.4 mg/L), but higher than in the deep sea.

These three SCE were observed up to 20 September, i.e., the first SCE exists for more than 10 days. The trajectories of these SCEs computed from the MODIS images are shown in Figure 9a. These SCEs move to the northwest from the cape and then spin in an anticyclonic direction (Figure 9a). Such trajectories are probably caused by the advection of SCEs by a larger mesoscale anticyclonic structure situated to the west of Crimea, that is observed in Figure 8a. This mesoscale feature looks like an area with increased TSM in the form of a vortex dipole located to the southwest of to the west of the Crimean coast (red arrows in Figure 7a,b). Finally, they go to the northwest Crimean coast, to the Eupatorian cape (33°E, 45.2°N) and dissipate.

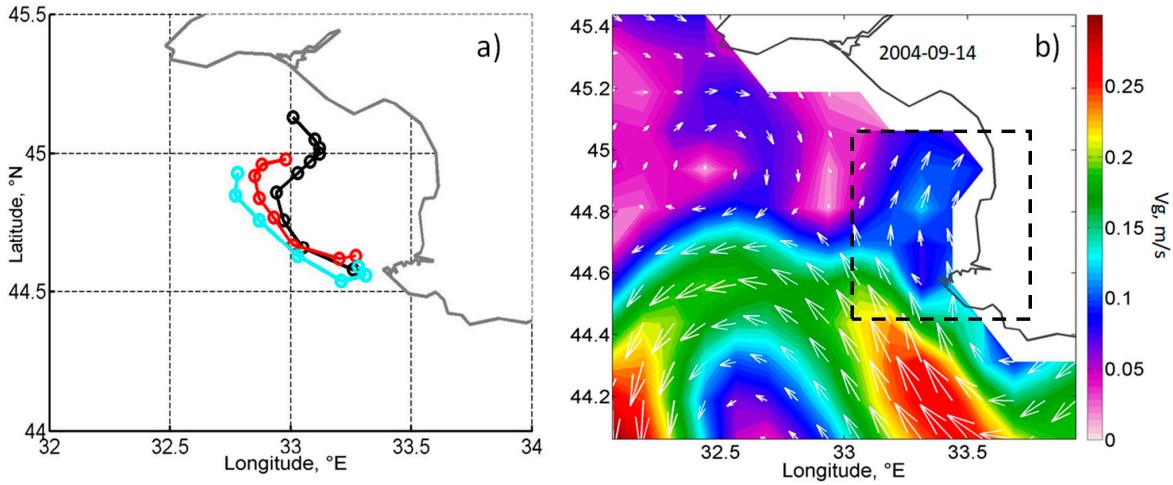

**Figure 9.** (**a**) The trajectory of the generated submesoscale eddies; (**b**) the distribution of the altimetry-derived geostrophic velocity on 14 September 2004.

The translational velocity of the two first SCEs was the highest (0.15 m/s–0.25 m/s) in the initial period of the eddies' formation and then decreased to 0.05–0.01 m/s (Figure 10a). These features of translational velocity are in agreement with the velocity of the background currents observed from the altimetry data (Figure 9b). Near the cape, the current turns clockwise and the velocity was higher 0.15–0.25 m/s than in the northern part shown in Figure 7b (0.1–0.15 m/s). This illustrates that the transfer of the observed SCE is mainly caused by their advection by the background currents.

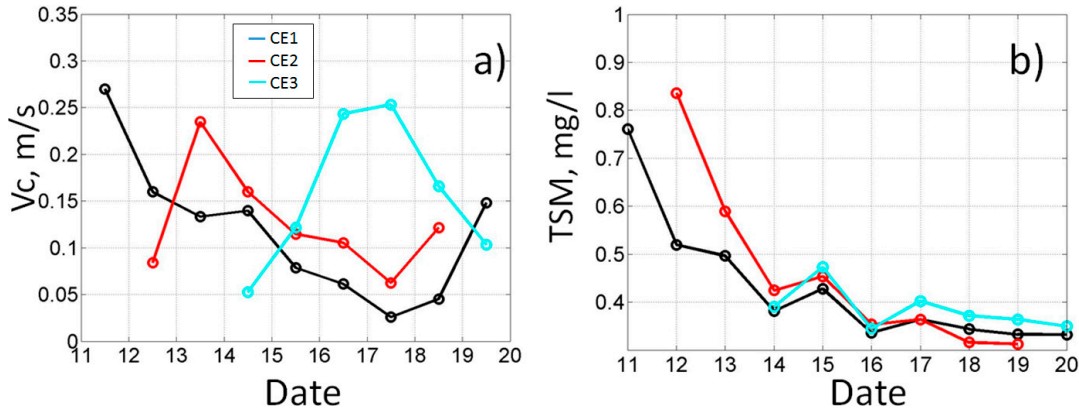

**Figure 10.** The evolution of (**a**) translational velocity and (**b**) TSM in the core of SCE from MODIS data on 11–20 September 2004.

The evolution of TSM in the SCEs core is shown in Figure 10b. TSM was maximal near the coast after the storm winds on 10 September 2004, and then began to decrease (Figure 5). That is why eddies C1 and C2 formed on 11 and 12 September, respectively, have an initial TSM = 0.8 mg/L, while eddy C3 formed later on 16 September has a lower TSM = 0.4 mg/L (Figure 10b). This indicates that TSM in SCE depends on the TSM in coastal waters, which these SCEs trap during the initial stage of their formation. We also note that the eddies were also characterized by increased concentration of chlorophyll A (1.5 mg/m$^3$ on 11 September 2004) as compared to the surrounding waters (not shown).

During eddies' evolution, TSM in their core noticeably decreases (Figure 10b). The rate of decrease was highest in the first stages of eddy's lifetime. In the first three days, the TSM in C1 and C2 fall down two times from 0.7–0.8 mg/L to 0.4–0.5 mg/L. During the next 6 days, TSM values in the eddy core decreased to 0.3 mg/L, which was still 1.5 times higher than the background values of TSM (0.2 mg/L). This indicates that turbid coastal waters trapped in the eddy core gradually mixes with clean waters of the deep sea.

### 3.3.3. Case 3. August 2003

The next example demonstrates the formation of a series of submesoscale eddies near the Crimean coast on the front of wind-driven upwelling on 14 August 2003. The small-scale structure of the upwelling is seen distinctly on the high-resolution image of Landsat −7 on 14 August 2003 (Figure 11). Upwelling expands to the west to Cape Hersones and then forms a thin elongated filament. Gradients of density on this filament cause the development of the frontal current directed offshore from Cape Hersones. This current traps the coastal matter from the coast and transports it to the deep sea, which is seen as a band of turbid waters in Landsat-7 optical data.

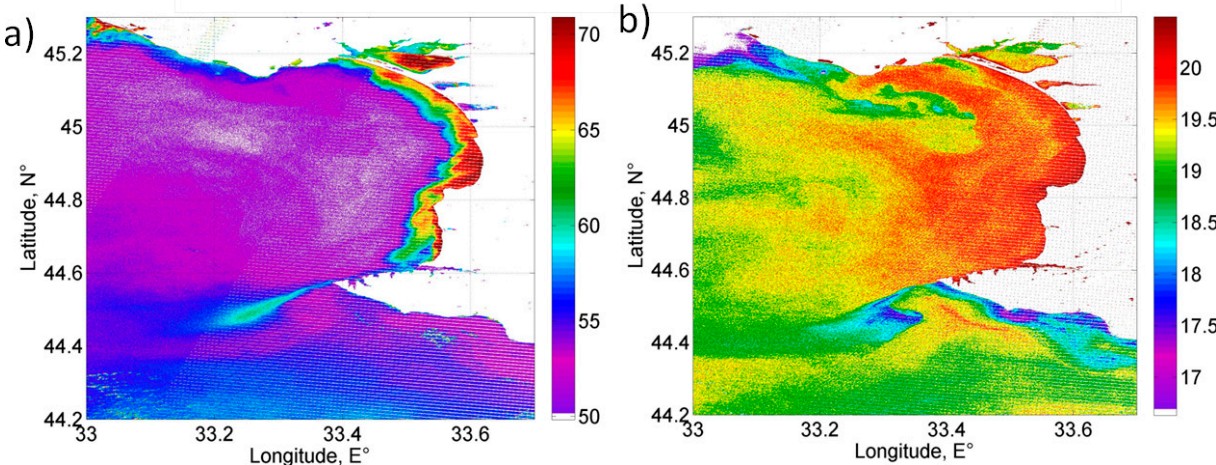

**Figure 11.** The frontal turbid jet on the periphery of upwelling on the Landsat-7 data for 14 August 2003: (**a**) optical radiance in band 1 (0.45–0.52 nm) (Watts/m$^2$/srad/μm); (**b**) temperature in infrared band 6 (°C).

The further evolution of this jet is observed in the series of MODIS images on 15–24 September 2003 in Figure 12. On 15 September, the formation of the spot with a high TSM (1 mg/L) is observed at the end of the offshore jet. The spot has a radius of 5 km. On 17 September, this spot detaches from the jet and moves ~17 km to the southwest. On the basis of the previous examples, we can suggest that this spot represents an SCE (C1). The spot can be seen as its TSM values (0.8 mg/L) is two times higher than the TSM of the surrounding waters (0.4 mg/L). The velocity of the SCE movement is 0.2 m/s, which closely corresponds to the geostrophic velocity of the Rim current from altimetry data.

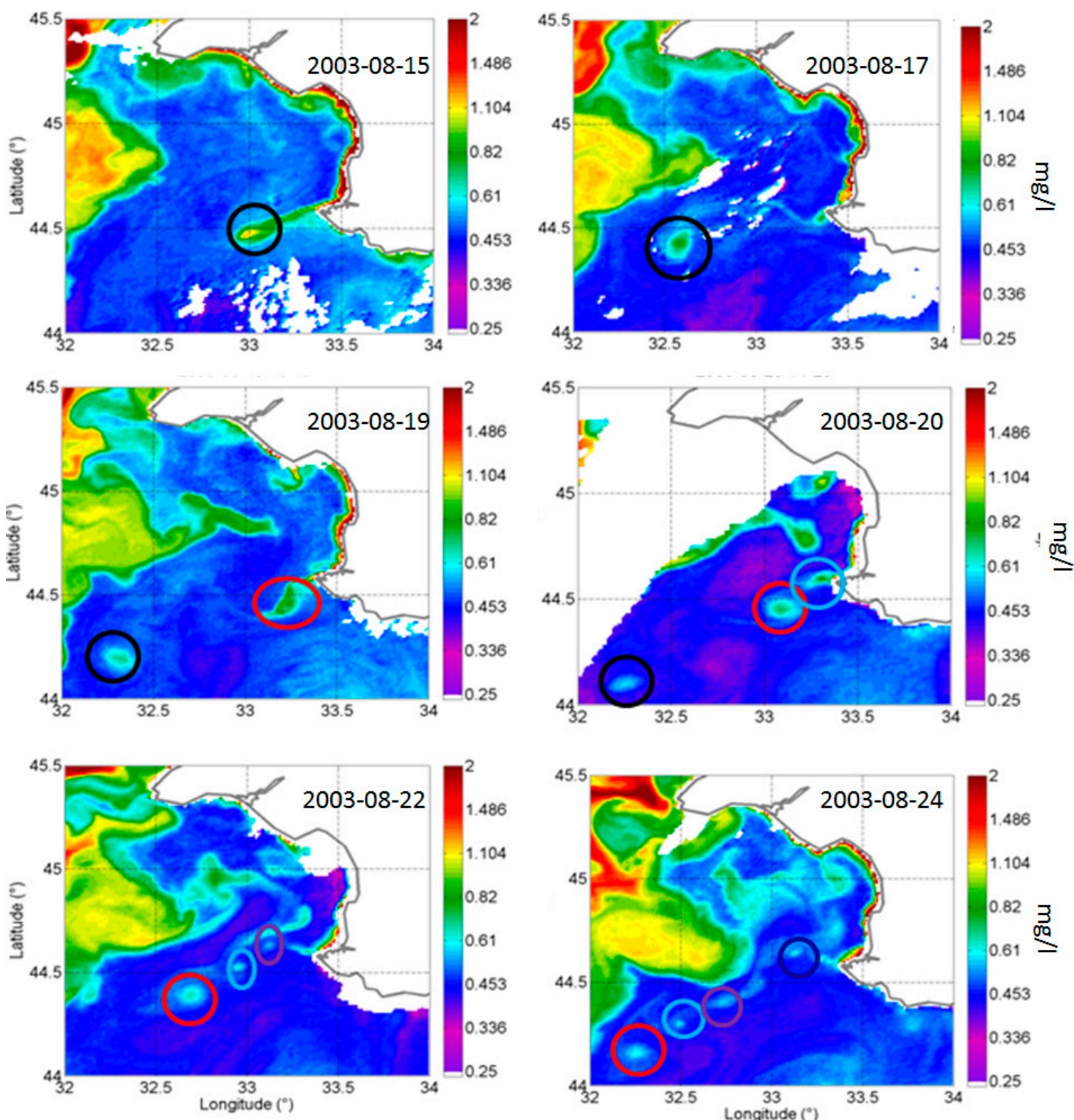

**Figure 12.** MODIS maps of TSM (mg/L) for 15–24 August 2003 showing the generation of a series of submesoscale SCE, their separation from the coast and transport of TSM in the deep part of the basin.

This eddy was also seen in the SST maps. SST in C1, generated on the front of upwelling, was 2° colder than the surrounding waters. During its lifetime, SST in C1 constantly rises (Figure 13) and finally reaches a value, which is similar to the deep sea values.

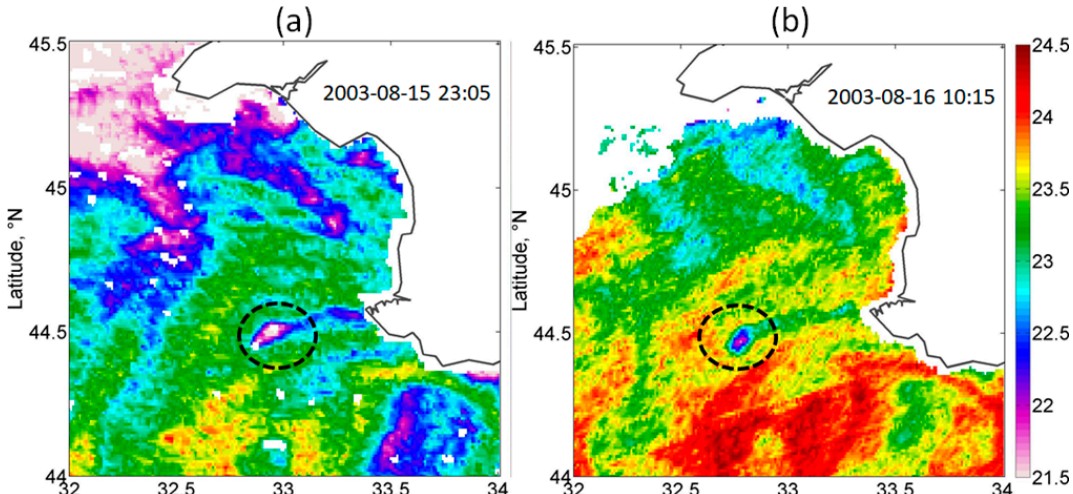

**Figure 13.** MODIS maps of sea surface temperature (°C) for 15 (**a**) and 16 (**b**) August 2008 showing the movement of a cold SCE from the coast.

On 19 August, another more prominent SCE (C2) formed near Cape Hersones. At this time, C1 further displaced to the southeast to the point 44.2°N, 32.4°E. On 20 September, another SCE (C3) with increased TSM (0.6 mg/L) is generated near the cape. C1 and C2 further displace to the southwest. Two more SCEs were generated on 22 and 24 September in the same area, respectively. Thereby, we observed a series of five submesoscale eddies formed near the cape on the front of upwelling. The radius of the first generated eddies (C1–C2) was larger than the last ones (C3–C5). All these eddies (C1–C5) trap TSM near the coast and further transport it to the southeast.

The translational velocity and direction of this propagation corresponds well to the altimetry-derived Rim current characteristics in this area. This suggests that the displacement of these eddies was mainly caused by their advection by the background currents. The eddies C1 and C2 during the time of their observation (5 days) were transferred to a distance of more than 100 km.

TSM in these SCEs at the time of their generation was close to the TSM values near the coast and then began to decrease (Figure 14). Particularly, TSM in C1 was about 1.1 mg/L on 15 September and further decreases to 0.6 mg/L on 20 September. Similar decrease in TSM from 0.8 mg/L to 0.6 mg/L is observed for the other SCEs. The decrease in SST and TSM anomaly in SCEs indicate that after the formation SCEs began to dissipate, and the trapped coastal waters gradual mixes with the surrounding clean sea.

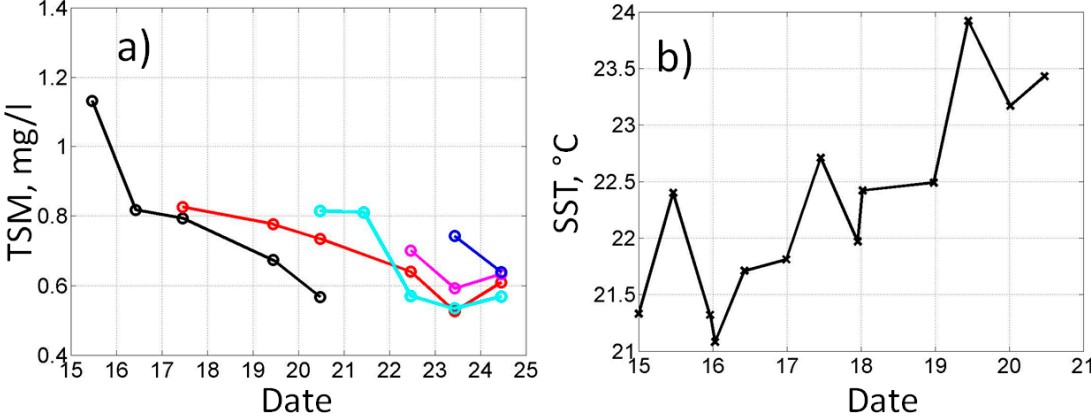

**Figure 14.** (**a**) The variability in TSM (mg/L) in the core of SCEs (different SCEs are shown by different colors); (**b**) the variability in surface temperature (°C) in the core of the first SCE from MODIS data on 11–20 September 2004.

Data on the radius and TSM in SCEs can be used to estimate their transport. In case 3, the eddy was formed at the end of the coastal jet, which develops on the shallow Crimean shelf with depths of 20 m. Then, we can estimate its thickness as h = 20 m. Similar estimates of SCE thickness in this area were obtained based on in situ data in [15]. Then, the volume of the trapped water will be V = h·pi·R² = 1.5 km³, where R is radius R = 5 km. Taking the anomaly of TSM in their core as 0.3 mg/L, it will contain an additional 450 tonn of suspended matter and transport it with a relatively fast velocity of 0.2 m/s.

### 3.4. Transport of Coastal Waters in SCE from NEMO Model Data

Small submesoscale eddies formed near the coast are reproduced with the numerical model NEMO with 1-km resolution. The example in Figure 15 represents the formation of two SCEs during the separation of coastal upwelling from the coast. Intense currents on the upwelling front cause its filamentation and elongation in the offshore direction. The cyclonic currents on the upwelling front transport warm waters, cut the filament and detach SCEs from the coast. These eddies are characterized by relatively low temperature and very high vorticity, which exceeds $10^{-4}$ 1/s, or 1f (Rossby number RO = 1), indicating their submesoscale dynamics (Figure 15).

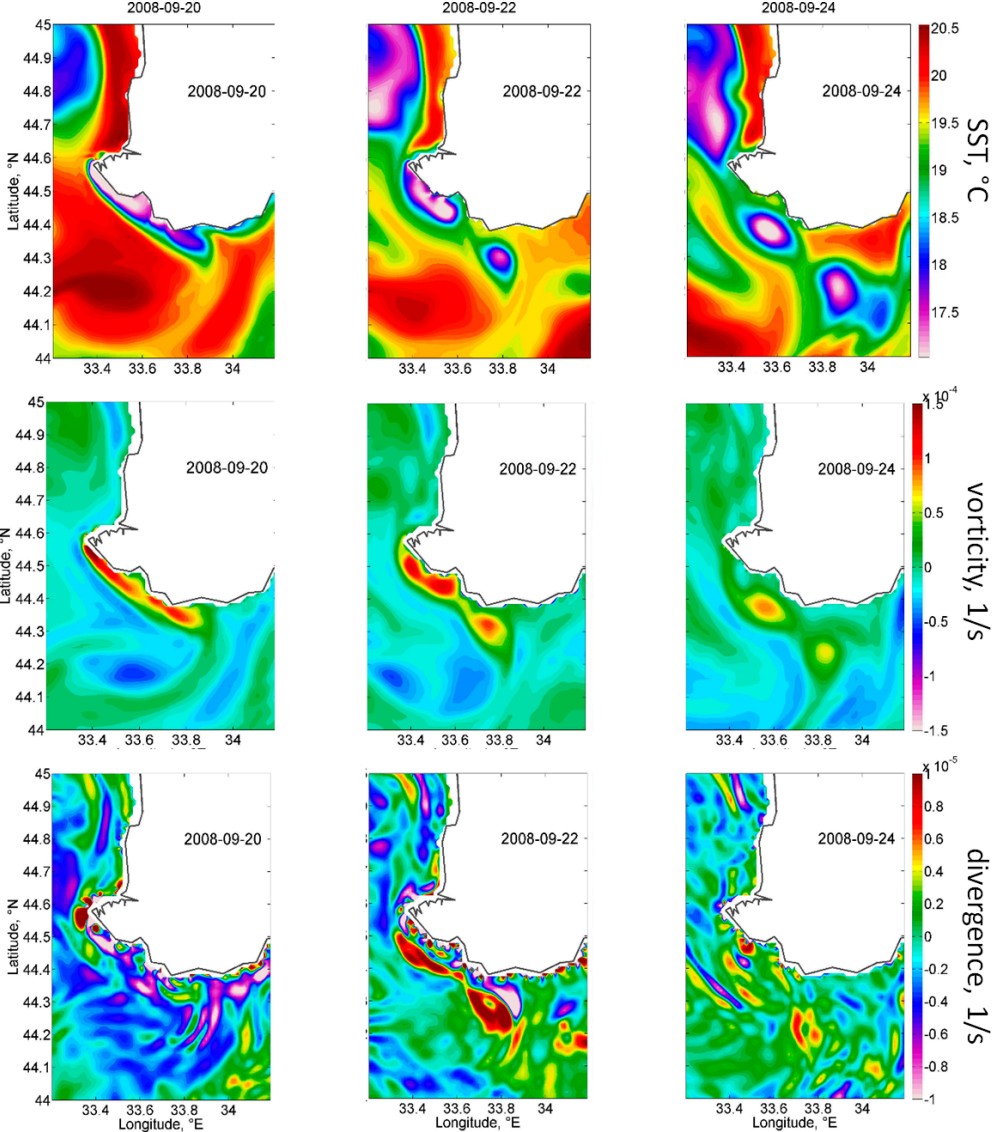

**Figure 15.** Top panels—temperature; middle—vorticity; bottom—divergence at z = 10 m, in the study area for 20 (**left**), 22 (**center**), 24 (**right**) September 2008, from NEMO model calculations.

Before the SCE formation, the study area is characterized by convergence ($div < 0$), indicating the weakening of the coastal upwelling (Figure 15—left). Then, during the eddy formation, absolute values of divergence reach maximum and have a characteristic shape similar to the satellite data: convergence in the coastal part of the eddy and divergence in its offshore part (Figure 15—middle). When SCEs detach from the coast, the convergence zone disappears, which shows the important impact of the coast on its formation. At this time, the SCEs are characterized by intense divergence on their periphery and near zero value of *div* in their center (Figure 15—right). Such a distribution is in agreement with theory, indicating a strong role of centrifugal forces on divergence in the "open-sea" SCE. This process may have caused the release of the accumulated TSM in the late stages of SCEs' life.

To analyze the effect of eddies on the transfer of coastal waters, 11,683 particles were released in the coastal part on 19 September 2008, a few days before the formation of eddies. They were released at each grid node in 0–100 m, i.e., about 360 at each horizon with a spatial step of 1 km (points in Figure 16b). Further, the three-dimensional trajectories of these particles were calculated using the Runge–Kutta scheme.

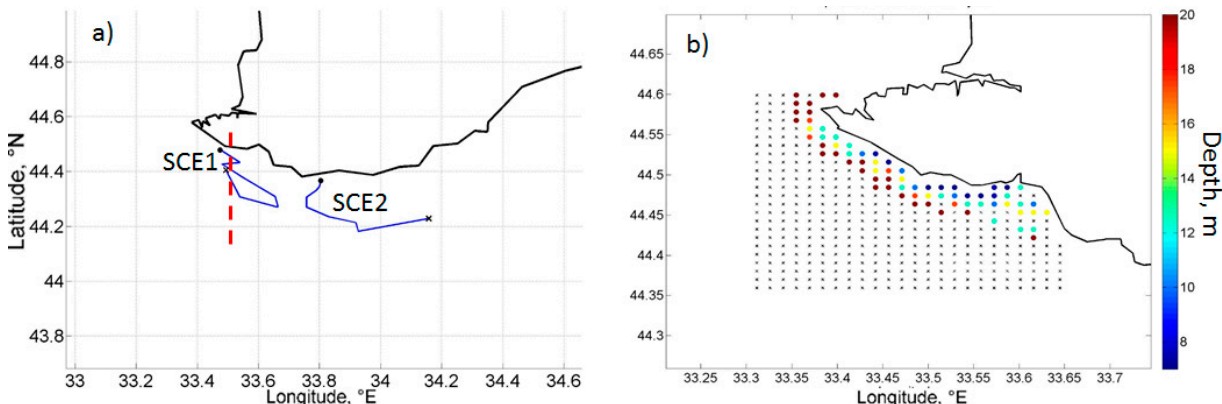

**Figure 16.** (**a**) Trajectories of identified eddies; (**b**) starting points of Lagrangian particles. The color shows the maximum initial depth of the particles that were trapped by the SCEs. Red dashed line shows the position of the section for Figure 17.

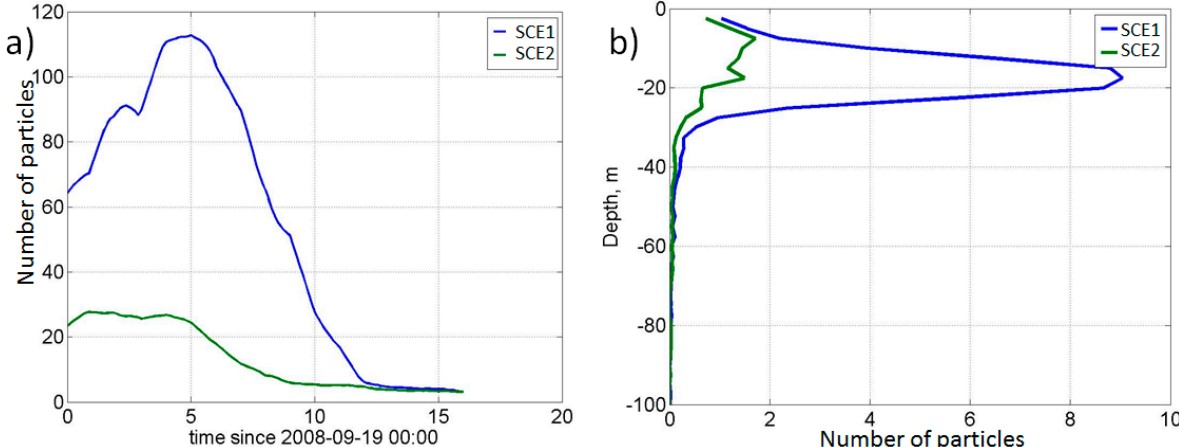

**Figure 17.** (**a**) Number of particles trapped by the western eddy SCE1 (blue line) and eastern SCE2 (green line); (**b**) time-averaged vertical distribution of the number of particles located in SCE1 (blue line) and SCE2 (green line).

Further, we analyze the particles that were situated in the SCE area determined via the automatic identification method. Their trajectories which were computed using methods

of automated identification are shown in Figure 16a. Initial position of the particles, that were trapped by SCE1 and SCE2 are marked in Figure 16b in color. In the coastal part of upwelling, eddies trap particles from the upper (0–10 m) layer; in the seaward part, particles located in the subsurface layer (10–20 m) are trapped.

The maximum number (110) of particles trapped by SCE1 (blue line) was 5 times higher than the maximum number of particles trapped by SCE2 (Figure 17). The sections of salinity, temperature and velocity in the center of SCE1 are presented in Figure 18. The eddy with a radius less than 10 km is clearly visible in the 0–30 m layer. Its orbital velocities reach rather high values, up to 0.4 m/s (Figure 18c). The rise of the thermo- and halocline was observed in the eddy core, leading to significant anomalies in salinity and temperature in the 0–50 m layer (Figure 18a,b).

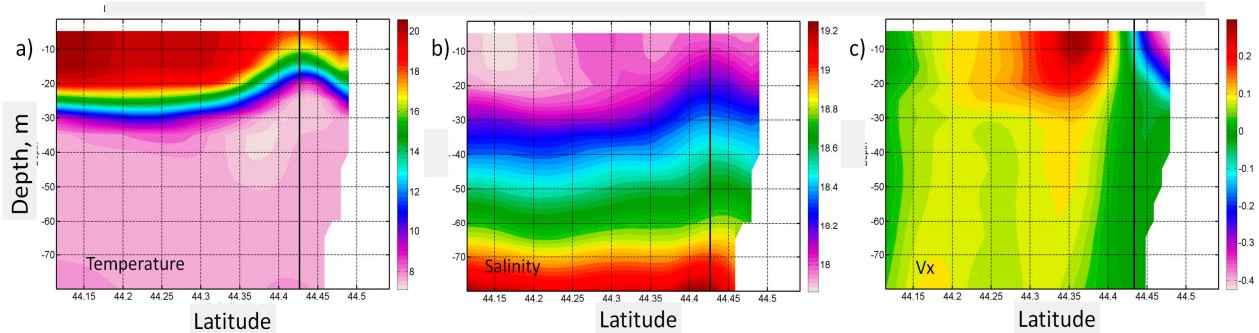

**Figure 18.** Sections of (a) salinity, (**b**) temperature, (**c**) zonal velocity across SCE1 on 23 September 2008. Black line shows the position of the eddy center.

Our analysis showed that the isotherms descend in the SCE during its evolution. For example, the isotherm 16° at the initial moment of SCE formation (September 20) was close to the surface. On September 23, it deepened to 10 m, and on September 27 to 16 m. This indicates a gradual attenuation of the eddy, which begins immediately after its separation from the coast.

Figure 19 shows the evolution of the divergence on the latitudinal section across the eddy center. In the initial period of eddy formation (Figure 19a), an area of intense convergence ($div < -3 \times 10^{-5}$ 1/s) is observed near the coast. It occupies the upper 0–35 m layer, which correspond to the eddy vertical size (Figure 18a). The width of this zone is ~2 km, about $\frac{1}{2}$ of eddy radius. On 23 September, the eddy moves offshore. At this time, the area of convergence is observed only in its lower part (Figure 19b), where the eddy touches the continental slope. On 24 September, the eddy separates from the coast and the convergence values decrease to minimal values (Figure 19c).

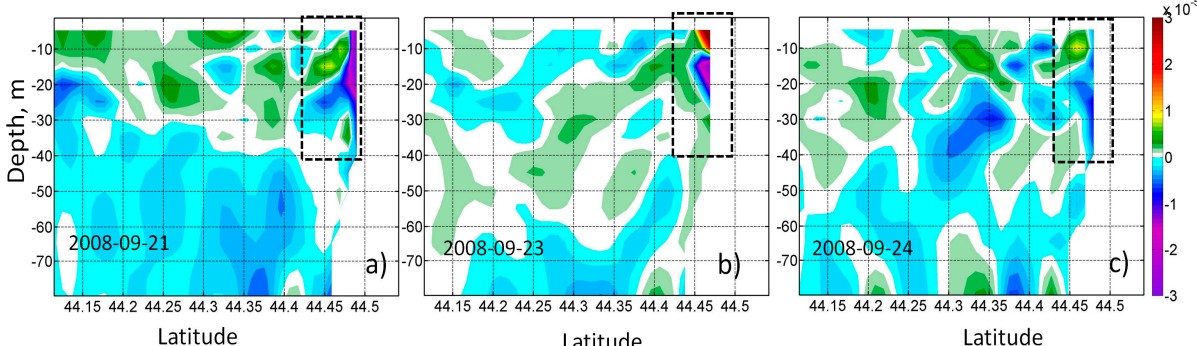

**Figure 19.** Latitudinal section of the divergence (1/s) across the eddy center on 21 September 2008 (**a**); 23 September 2008 (**b**); 24 September 2008 (**c**). Black rectangle shows the position of convergence near the coast.

The trajectories of particles that were trapped by SCE1 at different horizons are shown in Figure 20. As can be seen, the particles mostly made two or, in rare cases, three cycles around the SCE center in the upper layers. Then, after dissipation of the vortex, they began to move to the northwest along with the main Rim current. It is clearly seen that the radius of particle trajectories decreases in the deeper layers. This reflects the cone-shaped structure of the vortex with a large radius in the upper part (Figure 18c).

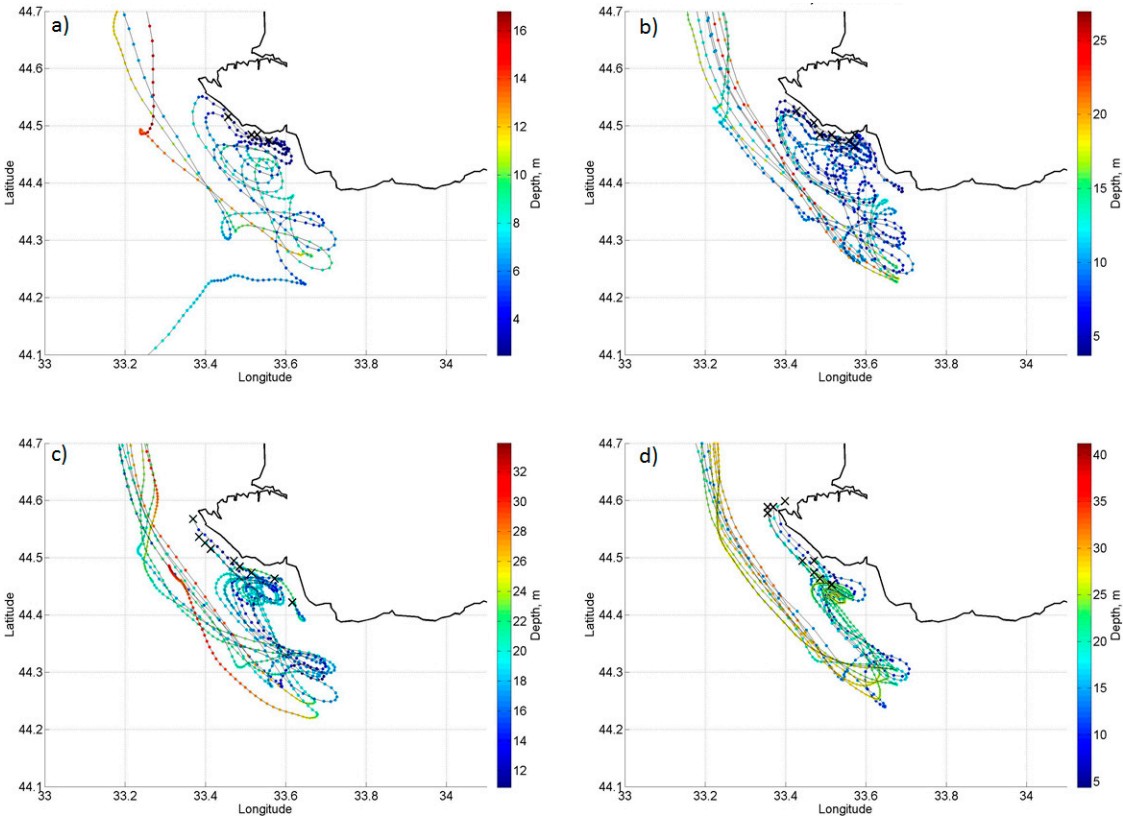

**Figure 20.** Examples of the trajectories of particles captured by the westerly vortex at different horizons, z = 2.5 m (**a**), 10 m (**b**), 20 m (**c**), 25 m (**d**).

The time-averaged profile of the number of particles in the eddy (Figure 16b) shows that most particles trapped in the eddy were located at depths of 0–30 m, which closely corresponds to its vertical position (Figure 18c). The maximum number of particles was trapped in the 15–25 m layer. These depths approximately corresponded to the position of the seasonal thermocline in the eddy, i.e., the vortex core with the maximum density gradients.

The number of particles in the western eddy in the initial period increases and reaches a maximum (110) on September 24, and then gradually decreases to 0 (Figure 17). This variability is associated with the evolution of the eddy. The vorticity of SCE1 reaches its maximum on the third day of the calculation, i.e., 23 September (Figure 21a). This happens when the eddy finally breaks away from the shore and enters the warm waters of the open sea. This process leads to an increase in horizontal density gradients along the entire periphery of the vortex, and the rise of its kinetic energy and vorticity. An intensifying vortex more effectively traps water in its core at depths of the seasonal thermocline. The maximum number of particles is observed with a delay of 1 day from the maximum vorticity. After that, the vorticity of SCE begins to gradually decrease and the isopycnals in the eddy begin to level-out. Due to the absence of external driving forces and gradual mixing with the waters of the open sea, the eddy dissipates.

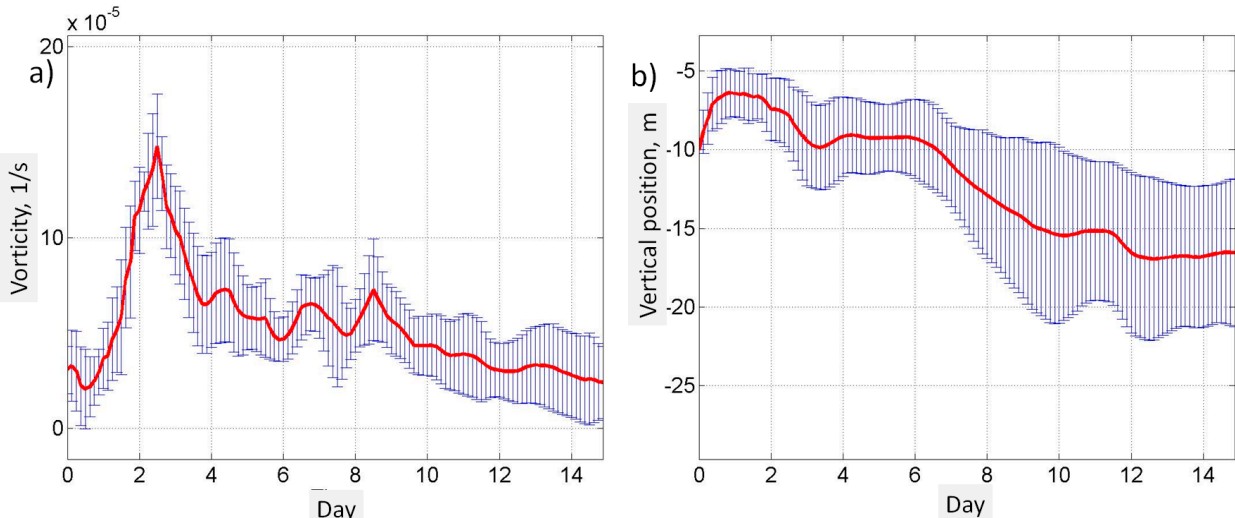

**Figure 21.** Time variability in the average (**a**) vorticity; (**b**) vertical position of the particles initially released at a depth of z = 10 m and trapped in the SCE. Red lines show the standard deviation.

This process is well-expressed on the graph of changes in the vertical position of particles in the vortex, initially located at a depth of 10 m (Figure 21b). In the initial period of eddy amplification, particles rise to depths of 6–7 m, and then begin to descend, reaching depths of 16–17 m by the end of the eddy's existence. Thus, SCEs not only cause the transfer of particles from the shore, but also contribute to their deepening into the lower layers, caused by the dissipation of SCEs.

Data on the number of trapped particles in the eddy make it possible to estimate its effect on the transport of coastal waters. The volume of one particle is $vi = 1 \text{ km} \times 1 \text{ km} \times 2.5 \text{ m} = 0.0025 \text{ km}^3$, and the maximum number of particles N = 110 corresponds to the total volume of 0.25 km$^3$.

## 4. Discussion

In this article, we investigate the impact of SCEs on the cross-shelf transport of the coastal waters. We show that small SCEs with radii less than 5 km are able to trap and accumulate suspended matter (or chlorophyll A) in their core and transfer it offshore to a distance exceeding hundred kilometers. Our analysis of long-term medium and high-resolution satellite data allow to highlight one of the important mechanisms of this process. We show that SCEs cause a convergence in their coastal part, where the waters move inward toward the core of the eddy. This effect can be explained by the impact of rigid boundary (coast), which closes the circulation in SCEs. Due to this effect, coastal SCEs are often seen as a spiral of turbid waters spinning inward in the core of the eddy. Most often in the Black Sea, SCEs are observed near the capes (Figure 1), where lateral shear plays an important role in their formation. However, vortex street consisting of similar SCEs can be observed in the open ocean, where their generation is related to the bottom drag near the continental slope [47]. We can expect that the continental slope plays the same role as the coast (become a sloped closed boundary or a "wall"). If SCEs reach the bottom, they can cause convergence and trap the fluid in their core. Such an example is presented in our previous paper (Figures 7 and 8 in [10]) where several SCEs in the vortex street accumulate suspended matter and transfer it from the cape, where they were generated. We can expect that in this case, the most intense pattern of convergence will be observed near the bottom (see Figure 19b), and such eddies will affect the transfer of bottom sediments. Also, it is possible that frontal jets can also play the role of such a "wall" (Figure 5), where vertical velocities will be directed downward.

It should be noted that in this paper we only discuss one of the important mechanisms of the accumulation of coastal waters in SCEs. Several other mechanisms can cause the rise

of TSM or Chl in SCEs. First, SCEs can be formed in turbid waters. Initially, it will contain these waters in the core, and then, transport them offshore. Second, intense upwelling of the deep nutrient-rich waters in SCEs may cause the rise of phytoplankton and the increase in Chl in SCEs. The latter effect was particularly described in oceans [48] and in the Black Sea [49]. However, in most of the observed cases (Figures 2–4, 6 and 8), the jet of turbid waters usually entrains in an SCE from its periphery. The core of SCEs contains more clean waters in the initial stage of their formation. On satellite images, Chl and TSM concentration are maximal during coastal SCE development and then decrease with time. As phytoplankton need some time to grow, we can assume that biological effects play less important role in the evolution of Chl in the discussed eddies.

Model data allow us to estimate the vertical motions of the particles in the coastal SCEs. Interestingly, the particles generally move downward in these cyclonic eddies. Partly, this effect is related to the phase of SCE evolution, which impact the sign of vertical velocity in the eddies [48]. SCE weakens after their separation from the coast. That is why the isopycnals contained within them descend and particles move downward. This effect can also play a role in downwelling observed in the SCEs and may cause a rapid transfer of TSM in the deep layers of the basin. Unfortunately, we only have model data for 2008–2009. In these years, there were no specific examples of SCE transport in satellite data. Particularly, in the case described in Section 3.4, the sea was covered by clouds and no MODIS data were available. However, we should mention that the model was able to capture the spatial structure, sizes of SCEs and the phase of their generation during some events with intense submesoscale dynamics, as seen in the satellite data [24]. Also, model data show the gradual attenuation of temperature anomalies in SCEs, which is in agreement with the satellite data presented in Figure 13.

In our previous study [10], we analyzed the statistics of the submesoscale eddy occurrence from high-resolution NEMO modeling data (see Figure 1). The probability of SCE detection is not high: only in certain places near the coast, it reaches 0.2, while in many others, it does not exceed 0.05. Many of these eddies are short-lived, i.e., they exist 1–3 days and do not separate from the coast. Our analysis of more than 5000 daily maps of MODIS for 2004–2020 and more than several hundred high-resolution Landsat and Sentinel-2 images showed that generation of coastal SCEs is mainly observed during upwelling or interaction of the mesoscale eddies with the capes [10]. Both these processes intensify in the Black Sea during the warm period of a year. Our ability to track SCEs is also somewhat limited by clouds and relatively low time resolution of high-resolution satellite data. This is why, on average, we can detect 2–3 good cases per year, where we can see a series of SCEs transporting TSM to a long distance, as described in this study. In comparison, 1–2 strong mesoscale eddies per year cause intense transport of the shelf waters in the western part of the Black Sea [32]. However, the volume of SCEs is significantly smaller than the transport caused by mesoscale eddies [31]. Therefore, the total impact of SCEs on cross-shelf exchange in the coastal zone is less important.

At the same time, SCEs may provide the fastest transport of the coastal matter to the deep layers of the basin. The described processes lead to short-time pulses of the different chemical and biological compounds in the inner parts of the ocean basins, which differ it from the "mesoscale" cross-shelf transport. We should note that the gradients of chemical or biological compounds can be very high between shelf and open sea waters [50]. That is why even a relatively small volume of coastal waters can have a significant impact on the deep sea ecosystem [51]. For example, SCEs can bring fish larvae or coastal phytoplankton species to a large distance away from their origin, which may lead to the observed diversity, seeding or invasion of biological species in different habitats. SCEs in this case, represent the islands of coastal waters in the deep part of the sea, analogous to mesoscale eddies in the deep part of the oceans.

Such SCEs may also play an important role in the erosion/abrasion of the coast. Particularly, they can cause resuspension of bottom sediments due to intense vertical shear,

and then, redistribute them in the shelf zone. To describe this effect, we need high-frequency data on the vertical distribution of optical properties, which is currently unavailable.

## 5. Conclusions

In this article, we use high-resolution satellite and numerical model data to describe the characteristics of small coastal SCEs (with radius less than 5 km) and their impact on the transport of coastal waters. The analysis show the following:

- Submesoscale cyclones accumulate coastal waters in their core and transport them offshore. Detailed analysis of their structure from satellite data shows that they represent a spiral, which spins inward in the core of SCEs. This indicates that SCEs are characterized by convergence and downwelling in the coastal part of their core, which contradicts the expected divergence and uplift observed in the cyclonic eddies. This contradiction can be explained by the impact of the "wall", which blocks the outward motion on the coastal periphery of the SCE and turn the water back to the core of the eddy. The hypothesis is confirmed through the analysis of the dynamic structure of SCEs from satellite and modeling data, which show the presence of the convergence pattern in the coastal part of SCEs and divergence in their offshore part;
- SCEs can transport suspended matter to a large distance away from the coast (~100 km). Often, series of SCEs are formed on the periphery of the mesoscale anticyclone or upwelling. Their separation from the coast cause a short-period entrainment of the coastal matter in the open ocean and the appearance of local anomalies in TSM and other optical properties. Such SCEs rapidly propagate offshore with a translational velocity of ~0.1–0.3 m/s due to the impact of background currents. The local anomalies of chlorophyll concentration, TSM and temperature in such SCEs gradually decrease during eddy's evolution but still can be observed for more than 1 week. The estimated transport of TSM in SCEs as per satellite data is about 50 tonn/s;
- Lagrangian analysis of high-resolution model data allows us to conduct an in-depth investigation of the transport of the coastal waters in the SCEs, which were formed in the area of coastal upwelling. The most intense entrainment of the particles is observed in the core of SCEs at the depth of the seasonal thermocline (15–25 m). This process intensifies when cold SCEs were separated from the coast and were surrounded by warm waters of the open sea. Then, the eddy began to gradually weaken. The particles made 2–3 rotations around its own center before the SCEs dissipated after ~10 days. Model data show that the decrease in SCEs' vorticity after the separation is accompanied by the flattening of initially risen isopycnals. Therefore, the coastal particles trapped in the SCEs rapidly deepen with time to 10–20 m, i.e., SCEs cause their downward transport. The total estimated volume of the trapped coastal waters from the Lagrangian analysis in one of these SCEs is 0.25 km$^3$.

The features of coastal SCEs discussed above distinguish them from the SCEs of the open sea, which cause intense upwelling and divergence of waters in their core [4,5]. The transport induced by coastal SCEs may cause intense local anomalies in the biological and chemical characteristics, and is another process that affects the functioning of the marine ecosystems.

**Author Contributions:** Conceptualization A.K. and S.S.; methodology A.A. and A.K.; software E.P. and A.M. (Artem Mizyuk); formal analysis A.A. and A.K.; investigation, A.A., A.K. and S.S.; resources E.P., A.M. (Artem Mizyuk), A.M. (Alesya Medvedeva), A.A.; data curation E.P. and A.A.; writing—original draft preparation A.K.; writing—review and editing A.A. and A.K.; visualization A.A. and A.K.; funding acquisition A.K. and S.S. All authors have read and agreed to the published version of the manuscript.

**Funding:** The evolution of submesoscale eddies were studied with the support of the RUSSIAN SCIENTIFIC FOUNDATION, grant number 21-77-10052. Satellite data acquisition and analysis was supported by state assignment No. FNNN-2021-0006, numerical modeling by state assignment No. FNNN-2021-0007.

**Data Availability Statement:** Satellite data is available from SentinelHub https://www.sentinel-hub.com/ (accessed on 23 July 2023) and OceanColor data archive (http://oceancolor.gsfc.nasa.gov//, accessed on 23 July 2023). The results of the numerical calculation for the study area are freely available at (https://zenodo.org/record/5607591, accessed on 23 July 2023).

**Conflicts of Interest:** The authors declare no conflict of interest.

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
