# Peer review of "Accumulation and Cross-Shelf Transport of Coastal Waters by Submesoscale Cyclones in the Black Sea"

_remotesensing, doi:10.3390/rs15184386_

Round 1

Reviewer 1 Report

This manuscript discusses the occurrence of submesoscale cyclonic eddies (SCE) in the coastal waters of the Black Sea and their impact on cross-shelf transport. The authors demonstrate that the presence of closed boundaries leads to convergence and accumulation of suspended matter within SCE, resulting in an inward spiral movement towards the core near the coast. 

Overall, the paper presents strong findings and provides a clear methodology and data description. The conclusions are convincing; however, there are some concerns that need to be addressed before publication.

Major concern:

1. The authors are encouraged to provide a detailed explanation of the distinctions between coastal SCE and the well-known topographic-induced vortex street  (as described in doi: 10.1002/2015GL063731) found along the continental slope. Focus on the generation mechanisms and clarify if coastal SCEs would be able to form or exhibit the ability to accumulate suspended matter without the close boundary blockage.

2. It is advised for the authors to avoid excessive listing of similar SCE examples in the manuscript, such as "two SCEs" and "three SCEs" in section 3.2 and "case 1," "case 2," "case 3," "c1-5" in section 3.3. Instead, the authors should select representative SCEs, emphasize their unique features, and provide comprehensive explanations.

3. Will tidal have any influence on SCEs?

4. The data assimilation based on the optical flow method needs to be clearly explained. Which observations are used for data assimilation? Is the data assimilation implemented with the same NEMO model? Is the submesoscale processes simulated by the ~1km free-run model? How much is gained by the data assimilation in term of the targeted submesoscale eddies?

5. As shown in Figure 3, can the divergence be captured by satellite reflectance data? What is its resolution? How is the velocity in Figure 3 computed? 

6. Usually the model cannot reproduce the exact submesoscale eddy corresponding to the observation, while this can be presented from a statistical point of view. An experiment with particle trajectories may therefore not be proper in drawing conclusions. 

General comment:

The language needs polishing. For example, single case and plural are sometimes mixed in one sentence. While I corrected some language errors, I did not pay full attention to these in all parts of the manuscript.

Minor comments:

L99 ‘WE’ should be 'We'

L115 ' (-4·107 m4/s) and (-8·106 m4/s)' ??? what's mean???

L129 '... suspended matter (TSM) it is seen that ...' should be '... suspended matter (TSM), it is seen that ...'

L150 'Their radiuses is...' should be ' Their radii are....'

L159 It would be nice if the the scale of the distance is marked in Fig.1

L165 'red rectangles' should be 'black rectangles'

L184 'ae' change to 'are'

L267 The space after the unit should be uniform

L381-384 A repetitive paragraph.

L397 'bet' change to 'be'

L499 Perhaps the authors should be compulsory to introduce error bars, confidence limits.

Author Response

This manuscript discusses the occurrence of submesoscale cyclonic eddies (SCE) in the coastal waters of the Black Sea and their impact on cross-shelf transport. The authors demonstrate that the presence of closed boundaries leads to convergence and accumulation of suspended matter within SCE, resulting in an inward spiral movement towards the core near the coast. 

 Overall, the paper presents strong findings and provides a clear methodology and data description. The conclusions are convincing; however, there are some concerns that need to be addressed before publication.

Response: At first, we would like to thank the reviewers for the useful comments, which help us to improve the manuscript

Major concern:

  1. The authors are encouraged to provide a detailed explanation of the distinctions between coastal SCE and the well-known topographic-induced vortex street (as described in doi: 10.1002/2015GL063731) found along the continental slope. Focus on the generation mechanisms and clarify if coastal SCEs would be able to form or exhibit the ability to accumulate suspended matter without the close boundary blockage.

Response: Generally, the continental slope plays the same role as the coast (become a sloped closed boundary or a “wall”). So, if SCE reaches the bottom, we can expect that the discussed effects will be also observed in vortex streets. Particularly, such an example is presented in our previous paper (Aleskerova et al., 2021, fig. 7 and 8). Zoomed image shown below demonstrates that SCE in a vortex street also accumulates and transports suspended matter in their core. We extend the discussion about this topic in the revised version of the paper according to your advice.

In the present manuscript, we show the example 5, where the role of a wall is played by the frontal jet. This example demonstrates that similar effects may be observed in open ocean. However, here we focusing on the transport of coastal waters, that is why the example in Figure 5 is given mainly for the discussion.

Figure R1. Vortex street observed near the Crimea on Sentinel-2 images on 6 July 2017 from (Aleskerova et al., 2021) (please see the attached file)

Aleskerova, A., Kubryakov, A., Stanichny, S., Medvedeva, A., Plotnikov, E., Mizyuk, A., & Verzhevskaia, L. (2021). Characteristics of topographic submesoscale eddies off the Crimea coast from high-resolution satellite optical measurements. Ocean Dynamics71(6-7), 655-677.

  1. It is advised for the authors to avoid excessive listing of similar SCE examples in the manuscript, such as "two SCEs" and "three SCEs" in section 3.2 and "case 1," "case 2," "case 3," "c1-5" in section 3.3. Instead, the authors should select representative SCEs, emphasize their unique features, and provide comprehensive explanations.

Response: We agree that such an extended explanation can be somewhat boring. However, we show several examples in the text to prove that the discussed process is systematic. For example,  in Section 2.2 we demonstrate that many similar SCEs can be observed during one episode. Two SCE in Figure 2c are remarkably different, because they are located in the open sea, and transfer TSM to the deep part of the basin.

In Section 3.3 we also illustrate that the series of 5 SCE (not one) causes the cross-shelf transport of the coastal waters during one episode.

  1. Will tidal have any influence on SCEs?

Response: It is commonly assumed that in the almost enclosed Black Sea the tides are very weak,  so this effect seems to be minor.

  1. The data assimilation based on the optical flow method needs to be clearly explained. Which observations are used for data assimilation? Is the data assimilation implemented with the same NEMO model? Is the submesoscale processes simulated by the ~1km free-run model? How much is gained by the data assimilation in term of the targeted submesoscale eddies?

Response: The figure 3 is based on the 4d-var assimilation method of two consecutive satellite images (Landsat-8 and Sentinel-2). This method is some kind of optical flow methods and is used to reconstruct the current velocities only from satellite data. The detailed explanation of this algorithm is presented in our previous studies:

  1. Kubryakov, A., Plotnikov, E., & Stanichny, S. Reconstructing large-and mesoscale dynamics in the Black Sea region from satellite imagery and altimetry data—a comparison of two methods. Remote Sensing 2018b, 10(2), 239.
  2. Aleskerova, A., Kubryakov, A., Stanichny, S., Medvedeva, A., Plotnikov, E., Mizyuk, A., Verzhevskaia, L. (2021). Characteristics of topographic submesoscale eddies off the Crimea coast from high-resolution satellite optical measurements. Ocean Dynamics 2021, 71(6), 655-677.

We rewrite the description of the method to make it more clear

Response: To estimate the dynamic characteristics of SCE from satellite data we used the optical flow method called 4-D variational assimilation algorithm [31]. The algorithm is based on the analysis of the normalized radiance of two quasi-synchronous images of Landsat-8 and Sentinel-2 obtained within time interval of several minutes. The resolution of the reconstructed velocity fields is ~100m. The details of the algorithm and data processing can be found in [28,30, 31]. 

The NEMO model uses no data assimilation.

  1. As shown in Figure 3, can the divergence be captured by satellite reflectance data? What is its resolution? How is the velocity in Figure 3 computed? 

Response: For the velocity computation, we use 4d-var assimilation method applied to two consecutive satellite images of Landsat-8 and Sentinel-2. (see the comment below). The resolution of the reconstructed velocity fields is comparable to the resolution of initial Landsat data (30m). However, usually we need to smooth fields with 3*3 moving average, so the real resolution is about 100m. We can expect that this resolution gives a possibility to estimate the divergence field in SCEs, as it is comparable to the usual width of frontal zones. (please see the discussion in Chelton, 2022)

Chelton, D. B., 2023: Estimation of Surface Current Divergence from Satellite Doppler Radar Scatterometer Measurements of Surface Ocean Velocity. J. Atmos. Oceanic Technol., https://doi.org/10.1175/JTECH-D-23-0052.1, in press.

  1. Usually the model cannot reproduce the exact submesoscale eddy corresponding to the observation, while this can be presented from a statistical point of view. An experiment with particle trajectories may therefore not be proper in drawing conclusions. 

Response: We agree with this comment. In this article, we use NEMO numerical data to provide more detailed investigation of the fate of the particles entrained in similar SCEs. We also use it to provide a description of the vertical structure of SCE and distribution of particles.

It is too difficult to estimate statistical data on the investigated process (cross-shore transport of TSM by SCE), because it is rather complex. It depends on the amount of TSM in the coastal zone, frequency of SCE generation, trajectories of SCE, which are related to the background currents e.t.c. However, according to Your comment, we added in the paper some statistical information about the probability of the SCE observation according to NEMO data. We also added Figure 1 to the study for the discussion of the total impact on such a process on the cross-shelf transport of coastal waters.

Comments on the Quality of English Language

General comment:

The language needs polishing. For example, single case and plural are sometimes mixed in one sentence. While I corrected some language errors, I did not pay full attention to these in all parts of the manuscript.

Response: Sorry for these mistakes. We carefully read the paper and corrected all the grammatical errors.

Minor comments:

L99 ‘WE’ should be 'We'

Response: Corrected

L115 ' (-4·107 m4/s) and (-8·106 m4/s)' ??? what's mean???

Response: Sorry. Corrected The corresponding values for turbulent viscosity and diffusivity are (-4·107 m4/s) and (-8·106 m4/s)

L129 '... suspended matter (TSM) it is seen that ...' should be '... suspended matter (TSM), it is seen that ...'

Response: Corrected

L150 'Their radiuses is...' should be ' Their radii are....'

Response: Corrected

L159 It would be nice if the the scale of the distance is marked in Fig.1

Response: For this task, we use the figure 2 , where a geographical scale is shown for the same eddies.

L165 'red rectangles' should be 'black rectangles'

Response: Sorry. Corrected.

L184 'ae' change to 'are'

Response: Corrected.

L267 The space after the unit should be uniform

Response: Corrected.

L381-384 A repetitive paragraph.

 Response: Sorry. Corrected

L397 'bet' change to 'be'

Response: Corrected

L499 Perhaps the authors should be compulsory to introduce error bars, confidence limits.

Response: Thank You for this good idea. We added error bars and redraw this figure.

Reviewer 2 Report

This paper discusses the impact of submesoscale eddies on the transport of suspended sediments in the Black Sea. Modern high resolution satellime observations allow to detect and follow the evolution of submesoscale eddies creating a great potential for research. This paper shows interesting examples of eddies development and proposes a mechanism that explains their features.

Unfortunately, this work left some questions unanswered.

The most important, how relevant are the studied events for the area? Some statistics of SCE appearance would be useful. This will also help to decide if the eddy-induced TSM transport is a significant constituent of the total TSM transport in the area. How do the 50tons of sediment transported by each eddy impact the local ecosystem?

The next concern is that the analysed events are separate examples, probably it would be better to properly compare them to each other, and to add some statistics describing the calculated eddies properties in different cases.

The numerical experiment also raises some doubts. Separate real events are shown, but then another event is modeled, which was not previously analyzed in the text. How well does the model represent the real event? The gap in the data in Figure 17 suggests that something may be wrong with the model.

The paper contain an excessive number of self-citations. 5-10% of self-citations is commonly asseptible.

Specific comments:

Line 34: Please explain what is f in "10f".

L34-36: "Such strong vorticity should intensify transport barrier between eddy core and surrounding waters, making them even more effective for the accumulation and transport of fluid in the ocean [7]"
This sentence is not very clear. How can the surrounding waters accumulate fluid?

L 47: Ref 15 uses satellite and laboratory, not in situ data.
Ref 22 "Vortex dynamics in the Southeastern Baltic Sea" does not discuss Black Sea.

L 54-63: The Introduction section needs a brief explanation of why the study of cross-shelf thansport of TSM is so important.

L 115-116: Please add reference to the values of turbulent viscosity and diffusivity.

l 142: The described eddies are in figures 1d and 2d.

L 153: The two larger eddies in Fig.2c look like spots, can they be used for obtaining the velocity field? In other words, how was it proved that they are both cyclonic?

L 175-176: It's Figure 3. Also, by 'coast' you mean a sharp depth gradient? There is clearly a bay on the map.

L 183-184: " Then, sometimes, as in an example in Figure 3c this process repeats. " What are the signs of this repetition?

L 186-189: The eddies in Fig.3b and 3c do not have a solid boundary, why is it concluded that its presence is the main reason?

L 205-209: If in Fig3b the negative div (big violet spot in the left) corresponds to the area where coastal waters are entrained in the eddy orbital motion, then this is an offshore current. So is there really a connection between an on/offshore direction and a sign of div?

Figure 4. What are the units of the color scale? What physical values are presented in figure 4b?

Figure 5: What do the letters A-E  and W represent?

Figure 6a needs axes or a grid, and a scale.

L 275-277: What is meant by "pulsating nature"? Where can we observe these pulsations?

L 296: "submesoscale SCE" = submesoscale submesoscale cyclonic eddy

L 309-310: "that is clearly observed in fig.7a." In Figure 7a I can see a big spot of increased TSM with may be 2-3 eddies. While in Figure 6a the MAE is observed clearly indeed.

Figure 8b: the subscription is wrong.

Figure 9: Eddies are denoted as C1, C2, C3 in the text and CE1, CE2, CE3 in the Figure.

Figure 10: Please add units of measurement to the scales.

Figure 11: Top and middle panels got mashed together.

L 372-376: This paragraph is repeated twice (lines 381-385).

L 413: Please add explanation to RO.

L 420-421: "absolute values of divergence reach maximum and have a characteristic shape similar to the satellite data"
If you are reproducing the real situation with NEMO, may be it would be useful to add the aforementioned satellite image.

L 431-432: " They were released at each grid node in 0-100m, i.e. 3600 at each horizon with a spatial step of 1 km (points in Figure 15b)." So how many horizons did you use?
12000 particles / 3600 = 3.333.

L 442: This "velocity" is a projection of velocity, judging from the figure. On what direction?

L 444-446: "The layer of high velocities coincides with the position of the seasonal thermocline, the upwelling of which is an important source of potential energy for this eddy (Figure 17a). The eddy has a significant effect on salinity and temperature, leading to a rise in the thermo- and halocline..."
There is contradiction in it. First you say that the eddy uses the energy of the upwelling, than that the rise of the thermocline is the effect of the eddy (so the energy of the eddy is used to lift the warm water mass).

Figure 17: The date in the subscription is August 13, 2008, while the previous analysis described the eddies appearing on September 20 and later.
Please show the position of this transsect on the map, in Figure 14. Why there is a blank in the data, if it is a model data? In Fig 17b a slight rise of isothermals can be seen surrounding the gap, what happened there?
Why the right part present in Fig 17a and 17b is missing in Fig.17c?

Figure 14 bottom center shows an interesting divergence pattern where it changes its sign. Why don't you make the trannsscet across this pattern? If you could show the upwelling and downwelling of isothermals corresponding to the convergence and divergence part of the pattern, it would strongly support the conclusion #1.

L 508-509 "In this article, we use high-resolution satellite and numerical model data to describe the characteristics of small coastal SCEs (with radius less than 5 km) and their impact on the transport of coastal waters."
You have estimated the transport of TSM with the eddies, but what role does this amount takes in total TSM transport? Without answering this question, the whole work does not meet the goal stated in the Introduction.

L 513-520 "This indicates that coastal SCE are characterized by convergence and downwelling in their core..."
"The hypothesis is confirmed by the analysis of the dynamic structure of SCE from satellite and modelling data, which show the presence of the convergence pattern in the coastal part of SCE and divergence in its offshore part;"
So where does the convergence occurs, in the core of the eddy  or in the coastal part?

L 529: "50 tonn·kg/s" - probably m/s.

Minor editing of English language is rrecommended.

Author Response

Response to Reviewer 2.

This paper discusses the impact of submesoscale eddies on the transport of suspended sediments in the Black Sea. Modern high resolution satellite observations allow to detect and follow the evolution of submesoscale eddies creating a great potential for research. This paper shows interesting examples of eddies development and proposes a mechanism that explains their features.

Unfortunately, this work left some questions unanswered.

Response: At first, we would like to thank the reviewers for the useful comments, which help us to improve the manuscript

The most important, how relevant are the studied events for the area? Some statistics of SCE appearance would be useful. This will also help to decide if the eddy-induced TSM transport is a significant constituent of the total TSM transport in the area. How do the 50tons of sediment transported by each eddy impact the local ecosystem?

Response: According to Your comment we add a discussion about the frequency of such events. We extend the discussion on this topic in the revised version of the manuscript

In our previous study (Aleskerova et al., 2021) we analyze the statistics of the submesoscale eddy occurrence from high-resolution Nemo modelling data (see figure 1 in the new version of the paper). The probability of SCE detection is not high: only in several places of the coast it reaches 0.2, and in many others it does not exceed 0.05. Many of these eddies are short-lived, i.e. they exist 1-3 days and do not separate from the coast. Our analysis of more than 5000 daily maps of MODIS for 2004-2020 and more than several hundred high-resolution Landsat and Sentinel-2 images showed that generation of coastal SCEs is mainly observed during upwelling or interaction of the mesoscale eddies with the capes [see also Aleskerova et al., 2021]. Both these processes intensify in the Black Sea in the warm period of a year. Our ability to track SCEs is also somewhat limited by clouds and relatively low time resolution of high-resolution satellite data. That is why, on average, we can detect 2-3 good cases per year, where we can see a series of SCE transporting TSM on the long distance, as it is described in this study. In comparison, 1-2 strong mesoscale eddies per year cause the intense transport of the shelf waters in the western part of the Black Sea (Kubryakov et al., 2018). However, the volume of SCE is significantly smaller than the transport caused by mesoscale eddies (see Shapiro et al., 2010). Thereby, the total impact of SCE on cross-shelf exchange in the coastal zone should be less important. At the same time, SCE may provide the fastest transport of the coastal matter to the deep layers of the basin. The described processes lead to the short-time pulses of the different chemical and biological compounds in the inner parts of the ocean basins, which differs it from the “mesoscale” cross-shelf transport. We should note that the gradients of chemical or biological compounds can be very high between shelf and open-sea waters. That is why  even a relatively small volume of coastal waters can have a significant impact on the deep sea ecosystem. For example, SCE can bring fish larvae or coastal phytoplankton species on large distance from their origin, which may lead to the observed diversity, seeding or invasion of biological species in different habitats. SCE in this case, represents the islands of coastal waters in the deep part of the sea, analogous to mesoscale eddies in the deep part of the oceans.

The next concern is that the analysed events are separate examples, probably it would be better to properly compare them to each other, and to add some statistics describing the calculated eddies properties in different cases.

Response: We added some information about the probability of such events to the paper (please, see the comment above). Unfortunately, it  is too difficult to estimate statistical data on the investigated process (cross-shore transport of TSM by SCE), because it is rather complex. It depends on the amount of TSM in coastal zone, frequency of SCE generation, trajectories of SCE, impacted by the background currents e.t.c. Also, the clouds often prevents the observation of the evolution of SCEs - we often see only the beginning of their formation or the end. That is why in this paper there were no goal to describe the statistics of these process. Instead, our task was to explain its physics and demonstrate its impact on the cross-shelf transport. To illustrate the similarity of the described process, we provide the description of three different examples in Section 5.

The numerical experiment also raises some doubts. Separate real events are shown, but then another event is modeled, which was not previously analyzed in the text. How well does the model represent the real event? The gap in the data in Figure 17 suggests that something may be wrong with the model.

Response: The gap in the data in this figure is actually the land where information is absent. The section was made through the south coast of Crimea. We corrected the figure to exclude such questions.

We agree that it will be better to use similar dates for the description of the SCE transport from model and satellite data. Unfortunately, we have model data only for 2008-2009. In these years, there were no beautiful examples of SCE transport. Particularly, during the case described in Section 6  the sea was covered by clouds and there were no MODIS data available. As our goal was to describe the impact of SCE on the cross-shelf transport we choose the best example from MODIS and model data to show it to reader.

However, we should mention that the model was able to capture the spatial structure and the phase of some events with intense submesoscale dynamics. As an example figure R4 below show the submesoscale eddies formed on the periphery of the upwelling near the Anatolian coast on 13 and 15 August 2009 (fig. R3) . We can see very intense submesoscale dynamics and very similar pattern on the surface temperature maps from model and MODIS data. Unfortunately, clouds obstruct our ability to track such eddies in this case.

Figure R3. Surface temperature during upwelling near the Anatolian coast from NEMO model data (top) and MODIS satellite measurements (bottom) at 13 (left) and 15 (right) August of 2009.

The paper contain an excessive number of self-citations. 5-10% of self-citations is commonly asseptible.

Response: We exclude several references to our studies and added several citations to the revised version of the manuscript.

Specific comments:

Line 34: Please explain what is f in "10f".

 Response: Corrected. f- Coriolis parameter.

L34-36: "Such strong vorticity should intensify transport barrier between eddy core and surrounding waters, making them even more effective for the accumulation and transport of fluid in the ocean [7]"
This sentence is not very clear. How can the surrounding waters accumulate fluid?

 Response: Thank You. Corrected. Such strong vorticity should intensify transport barrier between eddy core and surrounding waters, making SCEs even more effective for the accumulation and transport of fluid in the ocean.

L 47: Ref 15 uses satellite and laboratory, not in situ data.

 Response: Agree and corrected

Ref 22 "Vortex dynamics in the Southeastern Baltic Sea" does not discuss Black Sea.

 Response: We excluded this reference from the text

L 54-63: The Introduction section needs a brief explanation of why the study of cross-shelf thansport of TSM is so important.

 Response: We added such sentence to the text: The offshore transport of the shelf waters is one of the important sources of nutrients, biota for the Black Sea deep waters, and can significantly affect its salt balance

L 115-116: Please add reference to the values of turbulent viscosity and diffusivity.

 Response: We added the reference [37]

l 142: The described eddies are in figures 1d and 2d.

 Response: Thank You. Corrected.

L 153: The two larger eddies in Fig.2c look like spots, can they be used for obtaining the velocity field? In other words, how was it proved that they are both cyclonic?

 Response: Generally, the sign of eddies is estimated on the base of the tracers distribution in satellite image and is somewhat subjectiv . We are basing on our experience and make it manually We can see a tail in the south part of the eddy, which seems to be its beginning. Also, we are basing on the comparison with other eddies in the south coastal part of the scene, which are somewhat similar and are situated close to these two.

L 175-176: It's Figure 3. Also, by 'coast' you mean a sharp depth gradient? There is clearly a bay on the map.

 Response: Thank You for this comment. Definitely there is a Sevastopol bay. In fact, it is bounded by the large and thin pier, which is not seen from the satellite imagery. We agree that this seems somewhat misleading. We redraw this figure for clarity.

L 183-184: " Then, sometimes, as in an example in Figure 3c this process repeats. " What are the signs of this repetition?

 Response: We corrected the text for clarity: “Sometimes, as in an example in Figure 4c the jet moves inward along a spiral and then is blocked again by the previous spiral arm. So, this process repeats and the jet continue to move to the eddy center.”

L 186-189: The eddies in Fig.3b and 3c do not have a solid boundary, why is it concluded that its presence is the main reason?

 Response: The solid boundary is situated to the north of them (masked by white). We improve the figures 3b and 3c for clarity.

L 205-209: If in Fig3b the negative div (big violet spot in the left) corresponds to the area where coastal waters are entrained in the eddy orbital motion, then this is an offshore current. So is there really a connection between an on/offshore direction and a sign of div?

Response: Thank You for this valuable comment. We rewrite this paragraph for clarity. The negative values (convergence) are detected in the coastal part of the SCE, and positive values (divergence) on its offshore periphery. Another area of a strong convergence is observed in the area where coastal waters are entrained in the eddy orbital motion. Such eddies are mostly formed due to interaction of eastward along-shore currents with capes (see more details in). At the same time, cyclonic eddies induce the westward along-shore currents. The confluence of these currents is observed in the area of the current separation from the cape, where we observe the strongest convergence in figures 4b,c. This effect should promote the accumulation of TSM and can explain the maximal value of reflectance detected in this area of SCE on the Landsat images (Figure 3b-left).

It should be noted that the structure of the divergence of SCE also is affected by the presence of other factors, e.g. shear of background currents and the stage of evolution of these eddies (intensifying or decaying eddy), and can be very complex.

Figure 4. What are the units of the color scale? What physical values are presented in figure 4b?

Response: We add units to the figure caption (Watts/ m2/srad/μm), The figure 4b is also radiance in 5 channel.

Figure 5: What do the letters A-E and W represent?

 Response: We extend the description of the this figure  in the revised version of the manuscript and explain the meanings of all letters.

Figure 6a needs axes or a grid, and a scale.

Response: We added coordinates in fig. 6a.

 This figure shows some combination of several channels on Sentinel-2 satellite image, so its scale have no clear meaning.

L 275-277: What is meant by "pulsating nature"? Where can we observe these pulsations?

Response: We use the word pulsating because usually we observe several SCE, which periodically formed near the coast and transport TSM to the deep part of the basin. So it seems like a pulsation of TSM to the deep part of the basin.  We corrected this phrase to “short-period “.

L 296: "submesoscale SCE" = submesoscale submesoscale cyclonic eddy

 Response: Thank You. Corrected.

L 309-310: "that is clearly observed in fig.7a." In Figure 7a I can see a big spot of increased TSM with may be 2-3 eddies. While in Figure 6a the MAE is observed clearly indeed.

 Response: We agree that it is not so clear. We extend the description of this eddy. Below You can see the zoomed figures showing this eddy with red arrows highlighting tracers distribution. Such tracers distribution is a usual sign of a vortex dipole in satellite imagery. For the reader, we added a red arrows to these figures to illustrate the manifestation of this larger mesoscale eddy and corrected the text

“Such trajectories are probably caused by the advection of SCE by a larger mesoscale anticyclonic structure situated to the west of Crimea that is observed in fig. 8a. This mesoscale feature looks like an area with increased TSM in the form of a vortex dipole located to the southwest of to the west of Crimea coast (red arrows in fig. 8a, b).”

Figure R4Zoomed figure 7a and 7b showing the vortex dipole (red arrows) in the distribution of TSM near the Crimean peninsula

Figure 8b: the subscription is wrong.

 Response: Thank You. Corrected.

Figure 9: Eddies are denoted as C1, C2, C3 in the text and CE1, CE2, CE3 in the Figure.

 Response: Thank You. Corrected.

Figure 10: Please add units of measurement to the scales.

 Response: We added the units of measurements.

Figure 11: Top and middle panels got mashed together.

 Response: Thank You. Corrected.

L 372-376: This paragraph is repeated twice (lines 381-385).

 Response: Thank You. Corrected.

L 413: Please add explanation to RO.

 Response: We added the explanation. Rossby number RO.

L 420-421: "absolute values of divergence reach maximum and have a characteristic shape similar to the satellite data"
If you are reproducing the real situation with NEMO, may be it would be useful to add the aforementioned satellite image.

 Response: We agree that it will be better to use similar dates for the description of the SCE transport from model and satellite data. Unfortunately, we have model data only for 2008-2009. In these years there were no beautiful examples of SCE transport. Particularly, during the described in Section 6 case the area was covered by clouds and there were no MODIS data available. As our goal was to describe the impact of SCE on the cross-shelf transport we choose the best example from MODIS and model data to show it to reader.

However, we should mention that the model was able to capture the spatial structure and the phase of some events with intense submesoscale dynamics (please see the comment above)

L 431-432: " They were released at each grid node in 0-100m, i.e. 3600 at each horizon with a spatial step of 1 km (points in Figure 15b)." So how many horizons did you use?
12000 particles / 3600 = 3.333.

 Response: Sorry we corrected the numbers. To analyze the effect of eddies on the transfer of coastal waters, 11683 particles were released in the coastal part on September 19, 2008, a few days before the formation of eddies. They were released at each grid node in 0-100m, i.e. about 360 at each horizon with a spatial step of 1 km (points in Figure 16b).

 Note that the amount of particles at each horizons is not the same as we have varying bathymetry.

L 442: This "velocity" is a projection of velocity, judging from the figure. On what direction?

 Response: Thank You. You are right - it is zonal velocity. Corrected.

L 444-446: "The layer of high velocities coincides with the position of the seasonal thermocline, the upwelling of which is an important source of potential energy for this eddy (Figure 18a). The eddy has a significant effect on salinity and temperature, leading to a rise in the thermo- and halocline..."
There is contradiction in it. First you say that the eddy uses the energy of the upwelling, than that the rise of the thermocline is the effect of the eddy (so the energy of the eddy is used to lift the warm water mass).

 Response: We agree and corrected this phrase: The rise of the thermo- and halocline was observed in the eddy core, leading to the significant anomalies of salinity and temperature at 0-50 m layer (Figure 18a, b).

Figure 17: The date in the subscription is August 13, 2008, while the previous analysis described the eddies appearing on September 20 and later.

 Response: Response: Sorry, the correct date is September 23. Corrected.

Please show the position of this transsect on the map, in Figure 14. Why there is a blank in the data, if it is a model data? In Fig 17b a slight rise of isothermals can be seen surrounding the gap, what happened there?
Why the right part present in Fig 17a and 17b is missing in Fig.17c?

Response: According to Your comment, we added a transect on the map in fig. 16a (red dashed line). The gap in the data in Figure 17 is actually the land, where information is absent. In the previous version of the manuscript. the section was made through the south coast of Crimea. Also the figure 17c was made for a shorter section (coordinates were different). We corrected the figure to exclude this inaccuracy in the revised version of the manuscript.

Figure 14 bottom center shows an interesting divergence pattern where it changes its sign. Why don't you make the trannsscet across this pattern? If you could show the upwelling and downwelling of isothermals corresponding to the convergence and divergence part of the pattern, it would strongly support the conclusion #1.

 Response: Thank You for  a good advice. We added the plot of the section of divergence to the paper. We also describe the evolution of divergence, which helps us to confirm our results.

Figure 19. Latitudinal section of the divergence (1/s) across the eddy center at a) 21 September 2008; b) 23 September 2008; c) 24 September 2008. Black rectangle shows the position of convergence near the coast.

L 508-509 "In this article, we use high-resolution satellite and numerical model data to describe the characteristics of small coastal SCEs (with radius less than 5 km) and their impact on the transport of coastal waters."
You have estimated the transport of TSM with the eddies, but what role does this amount takes in total TSM transport? Without answering this question, the whole work does not meet the goal stated in the Introduction.

 Response: According to Your comment,  we add a discussion about the frequency of such events. We extend the discussion on this topic in the revised version of the manuscript (please see comment 1)

L 513-520 "This indicates that coastal SCE are characterized by convergence and downwelling in their core..."
"The hypothesis is confirmed by the analysis of the dynamic structure of SCE from satellite and modelling data, which shows the presence of the convergence pattern in the coastal part of SCE and divergence in its offshore part;"
So where does the convergence occurs, in the core of the eddy or in the coastal part?
 Response:  We agree with this comment and corrected this phrase: This indicates that coastal SCE are characterized by convergence and downwelling in the coastal part of their core, which contradict expected divergence and uplift in the cyclonic eddies.

L 529: "50 tonn·kg/s" - probably m/s.

 Response: Corrected -  tonn/s

Reviewer 3 Report

The paper describes satellite observations of sub-mesoscale eddies in the Black Sea and results from a high-resolution numerical model. The aim is to assess the impact of sub-mesoscale eddies on cross-shelf exchange. The authors describe a few cases when sub-mesoscale eddies appear in satellite images of various water properties and speculate on the physical mechanisms responsible for the transport of these properties. Some quantitative estimates, although very few, are made. The study is aided by results from a numerical simulation of these processes in a high-resolution model. The main conclusion is that sub-mesoscale eddies formed in the coastal area in the Black Sea may provide an effective long-distance transport of coastal waters. From this perspective, the results might be interesting and worthy of publication. However, I have some concerns regarding both the clarity of the writing and interpretation of the results, for which I would request a substantial revision.    

 General:

The paper is a bit lengthy and contains long descriptions of satellite images where sub-mesoscale eddies appear as local anomalies of various water properties. The descriptions provide lots of irrelevant information and it is not immediately clear to me if they intend to illustrate a particular one process/phenomenon or they are different and illustrate different mechanisms. It is also not clear if these examples are typical or illustrate rather rare cases of long-lived coherent sub-mesoscale eddies.

The images themselves are difficult to read. Some (e.g. Fig 7) have a coastline explicitly shown but others (e.g. Fig 2, 3, etc.) have no such information and the reader has to guess where the coastline is. The authors somehow assume that the reader is familiar with the local geography and can quickly guess by the shape of a white spot in the image that this is Crimea peninsula or part of it. The same is true about geographic names which appear in the text (e.g., cape Hersones, p 290) but not on the map or image. Likewise, the paper would benefit from an introductory figure showing the Black Sea basin with relevant geographic features/ names, and, perhaps, schematically or as a mean field, the Black Sea circulation. Then the reader would know what and where the Rim Current is as well as the location of quasi-permanent mesoscale eddies mentioned in the text.     

The authors argue that the primary mechanism by which sub-mesoscale eddies trap coastal water is that related to its interaction with a “wall” (coast) or a narrow jet. Although this mechanism can contribute to a complex pattern of vertical velocity (or convergence/divergence) typically observed in sub-mesoscale eddies, this may not be the primary mechanism for water trapping. Sub-mesoscale eddies are highly nonlinear and trap water where they are formed, e.g. an eddy formed by baroclinic instability of a parent jet and detaching from the jet during the non-linear phase would have water of the parent jet (filament) trapped inside.  The spiral pattern typically observed in sub-mesoscale eddies is not necessarily related to the proposed mechanism but may be related to other phenomena (e.g., edge waves, interaction with internal waves, etc.).

Speaking of the proposed mechanism, in the schematic in Fig. 5, the patch of downwelling caused by hitting the wall (coast) is on periphery of the eddy. The eddy core is still characterized by upwelling, which makes sense. This is also what I see in Fig. 3 – general upwelling (div>0) around the eddy center (approximately), although the patterns are indeed very complex. In this regard, claiming that the observed sub-mesoscale eddies are somehow “totally” different from their open sea counterparts (conclusions, line 542) is misleading.

Minor:

Into line 18: “pulsating” – what does that mean???

Line: 34, 49 and elsewhere in the text: “10f” - ??? ‘f’ has never been introduced in the text.

Line 91: “RAS” – what is this???

Line 143: ‘its’ -??? (this is the first sentence in a new paragraph).

Line 138: “Then…” These are static images, they cannot demonstrate processes.

Line 164: ‘…by red rectangles’. The rectangles are black in fig. 2a.

Line 165: “Convergence…” I do not understand this sentence.

Line 174, fig.3: (b) and (c) – same dates (is it the same eddy?)

Line 179: “Initially,…”. Fig3a is a static image, it cannot show a process. All the discussion is thus a speculation.

Line 187: “closed boundary”-??? Rigid???

Line 188: “liquid continuity” -??? Fluid?

Line 208: “stipe of the eddy” – what is “stipe”?

Lines 209-211: “Such a convergence…” the sentence I don’t understand.

Line 212: “stipe” -???

Lines 214-218: does this example support or reject the hypothesis?

Line 225: change in ‘the orientation’ of the coastline

Line 226: “Then…” Fig 4 is a static image. It cannot show a process. All the discussion here is pure speculation.

Line 239: how the eddy “core” is defined?

Fig 5: what are the letters (in black boxes) in the figure?

Case1: Chl is not a passive tracer. The reduction in Chl concentration in the eddy does not necessarily mean mixing with surrounding waters and/or eddy weakening.

Line 289: “northeast storm”- ? northeast wind?

Line 291: “its headland”-???

Line: “…is clearly observed…” how? What is the signature of this eddy that we can clearly observe???

Fig 8b: I do not see any anticyclonic eddy (clearly) in the velocity map.

Fig. 9a: why is it relevant? It does not show any comparison with the background currents.

Line 313: “ the variability” – the distribution.

Line 338: I do not understand how the “rate of mixing” may depend on the gradient between the eddy and surrounding waters.

Lines 381-385: delete – repetition.

Line 396: “bet” – be

Line 483: contradicts to line 453 saying that eddies start to decay immediately after formation.

Line 523: “pulsating” – what does that mean?

The paper would strongly benefit from the careful proofreading. It is not practical to comment on all the details at this point; I will highlight just a few instances:

‘in’ satellite images (line 40)

diffusion ‘of’ momentum (line 114)

“a spots” (line 153). Spots – plural, ‘a’ is not needed.

‘motions (plural)…spin…’ (line 155); common mistake in the text

Careful proofreading is needed. There are multiple gramma errors, including  inconsistent use of verb tenses, single/plural, in/on, etc. 

Author Response

Response to Reviewer 3

The paper describes satellite observations of sub-mesoscale eddies in the Black Sea and results from a high-resolution numerical model. The aim is to assess the impact of sub-mesoscale eddies on cross-shelf exchange. The authors describe a few cases when sub-mesoscale eddies appear in satellite images of various water properties and speculate on the physical mechanisms responsible for the transport of these properties. Some quantitative estimates, although very few, are made. The study is aided by results from a numerical simulation of these processes in a high-resolution model. The main conclusion is that sub-mesoscale eddies formed in the coastal area in the Black Sea may provide an effective long-distance transport of coastal waters. From this perspective, the results might be interesting and worthy of publication. However, I have some concerns regarding both the clarity of the writing and interpretation of the results, for which I would request a substantial revision. 

Response: At first, we would like to thank the reviewers for the useful comments, which help us to improve the manuscript

 General:

The paper is a bit lengthy and contains long descriptions of satellite images where sub-mesoscale eddies appear as local anomalies of various water properties. The descriptions provide lots of irrelevant information and it is not immediately clear to me if they intend to illustrate a particular one process/phenomenon or they are different and illustrate different mechanisms. It is also not clear if these examples are typical or illustrate rather rare cases of long-lived coherent sub-mesoscale eddies.

Response: “ According to Your comment, we extend the discussion on this topic in the revised version of the manuscript:

“In our previous study (Aleskerova et al., 2021) we analyze the statistics of the sub-mesoscale eddy occurrence from high-resolution Nemo modelling data (see Figure 1). The probability of SCE detection is not high: only in several places of the coast it reaches 0.2, and in many others it does not exceed 0.05. Many of these eddies are short-lived, i.e. they exist 1-3 days and do not separate from the coast. Our analysis of more than 5000 daily maps of MODIS for 2004-2020 and more than several hundred high-resolution Landsat and Sentinel-2 images showed that generation of coastal SCEs is mainly observed during upwelling or interaction of the mesoscale eddies with the capes [see also Aleskerova et al., 2021]. Both these processes intensify in the Black Sea in the warm period of a year. Our ability to track SCEs is also somewhat limited by clouds and relatively low time resolution of high-resolution satellite data. That is why, on average, we can detect 2-3 good cases per year, where we can see a series of SCE transporting TSM on the long distance, as it is de-scribed in this study. In comparison, 1-2 strong mesoscale eddies per year cause the in-tense transport of the shelf waters in the western part of the Black Sea (Kubryakov et al., 2018). However, the volume of SCE is significantly smaller than the transport caused by mesoscale eddies (see Shapiro et al., 2010). Thereby, the total impact of SCE on cross-shelf exchange in the coastal zone should be less important. At the same time, SCE may provide the fastest transport of the coastal matter to the deep layers of the basin. The described processes lead to the short-time pulses of the different chemical and biological com-pounds in the inner parts of the ocean basins, which differs it from the “mesoscale” cross-shelf transport.”

The images themselves are difficult to read. Some (e.g. Fig 7) have a coastline explicitly shown but others (e.g. Fig 2, 3, etc.) have no such information and the reader has to guess where the coastline is. The authors somehow assume that the reader is familiar with the local geography and can quickly guess by the shape of a white spot in the image that this is Crimea peninsula or part of it. The same is true about geographic names which appear in the text (e.g., cape Hersones, p 290) but not on the map or image. Likewise, the paper would benefit from an introductory figure showing the Black Sea basin with relevant geographic features/ names, and, perhaps, schematically or as a mean field, the Black Sea circulation. Then the reader would know what and where the Rim Current is as well as the location of quasi-permanent mesoscale eddies mentioned in the text.   

Response: We agree with this comment and add the figure 1 to the text, which shows the main features of the Black Sea. The coastline is shown by the black line in figure 3. It is also present on all other figures except high-resolution images in figure 4, 5 and 6a. Our coastline (1km) is relatively coarse compare to these detailed images with 10-30-m resolution. That is why, it will spoil the quality of the visualization and we did not use it in this case. We also improve the figure 4 to make it more clear to the reader.

Figure 1. The probability of the identification of eddy with radius <10 km on the base of NEMO numerical model data in the Black Sea (a) and zoomed on Crimean coast from [Aleskerova et al., 2021].

The authors argue that the primary mechanism by which sub-mesoscale eddies trap coastal water is that related to its interaction with a “wall” (coast) or a narrow jet. Although this mechanism can contribute to a complex pattern of vertical velocity (or convergence/divergence) typically observed in sub-mesoscale eddies, this may not be the primary mechanism for water trapping. Sub-mesoscale eddies are highly nonlinear and trap water where they are formed, e.g. an eddy formed by baroclinic instability of a parent jet and detaching from the jet during the non-linear phase would have water of the parent jet (filament) trapped inside. The spiral pattern typically observed in sub-mesoscale eddies is not necessarily related to the proposed mechanism but may be related to other phenomena (e.g., edge waves, interaction with internal waves, etc.).

Response: We agree with this comment. Generally, the mechanism proposed in Your respectful comment is observed in figure 5. The eddy formed by baroclinic instability of a parent jet has water of the parent jet (filament) trapped inside. Here, we discuss mainly the reasons of the accumulation of the parent waters in SCE. Intense SCE should cause an upwelling and the rise of deep clean water in its core., while radial motions should displace the turbid waters to SCE periphery. However, on satellite images (e.g.. Figure 4) we see opposite effect – turbid waters eventually entrains in SCE core.

We agree that in our study we discuss here only one of the possible mechanisms. At least  two other mechanisms can exist. We revise text and write more accurately the discussion:

It should be noted that we discuss here only one of the important mechanisms of the accumulation of coastal waters in SCE. Several other mechanisms can cause the rise of TSM or Chl in SCEs. First, SCE can be formed in turbid waters. Initially, it will contain these waters in the core and, then, transport them offshore. Second, intense upwelling of the deep nutrient-rich waters in SCE may cause the rise of phytoplankton and the increase of Chl in SCE. The latter effect was particularly described in the ocean [Siegel et al., 2011] and in the Black Sea [Stanichny et al., 2021].

We should mention that our conclusions were based on the analysis of the large number of satellite measurements, where we often see the similar features, which gives some advantage to the discussed here mechanism. The jet of turbid waters usually entrains in SCE from its periphery, and the core of SCE contains clean waters in the initial stage of their formation. On satellite images , Chl and TSM concentration are maximal during eddy development in coastal SCE and then decrease with time. As phytoplankton need some time to grow, then we can assume that biological effect play less important role in the evolution of Chl in the discussed eddies.

 Speaking of the proposed mechanism, in the schematic in Fig. 5, the patch of downwelling caused by hitting the wall (coast) is on periphery of the eddy. The eddy core is still characterized by upwelling, which makes sense. This is also what I see in Fig. 3 – general upwelling (div>0) around the eddy center (approximately), although the patterns are indeed very complex. In this regard, claiming that the observed sub-mesoscale eddies are somehow “totally” different from their open sea counterparts (conclusions, line 542) is misleading.

Response: We agree that this part of conclusions was inaccurate and exclude “totally” from the text. We also rewrite the conclusions: . This indicates that SCE is characterized by convergence and downwelling in the coastal part of their core. We also extend the description of scheme in figure 6 to be more clear. Particularly, we point on the existence of divergence in the offshore part of SCE.

Minor:

 Into line 18: “pulsating” – what does that mean???

Response: We use the word pulsating because usually we observe several SCE, which periodically formed near the coast and transport TSM to the deep part of the basin. So it seems like a pulsation of TSM to the deep part of the basin. We agree that this phrase is not clear. We corrected it on “short-period “.

Line: 34, 49 and elsewhere in the text: “10f” - ??? ‘f’ has never been introduced in the text.

 Response: Corrected. f- Coriolis parameter.

Line 91: “RAS” – what is this???

Response: Sorry. Corrected. P. P. Shirshov Institute of Oceanology of the Russian Academy of Science

 Line 143: ‘its’ -??? (this is the first sentence in a new paragraph).

Response: Corrected: Two largest SCEs (Figure 1b) are situated in the western part of MAE.

Line 138: “Then…” These are static images, they cannot demonstrate processes.

Response: We agree and exclude this word from the text.

Line 164: ‘…by red rectangles’. The rectangles are black in fig. 2a.

Response: Corrected

Line 165: “Convergence…” I do not understand this sentence.

Response: We rephrase the Section title Convergence in the coastal part of submesoscale cyclones

 Line 174, fig.3: (b) and (c) – same dates (is it the same eddy?)

 Response: These are two different eddies observed on the same date. They have different coordinates.

Line 179: “Initially,…”. Fig3a is a static image, it cannot show a process. All the discussion is thus a speculation.

Response: Corrected throughout the text.

Line 187: “closed boundary”-??? Rigid???

Response: Corrected

Line 188: “liquid continuity” -??? Fluid?

Response: Thank You. Corrected.

Line 208: “stipe of the eddy” – what is “stipe”?

Response: Excluded from the text

Lines 209-211: “Such a convergence…” the sentence I don’t understand.

 Response: We rewrite this paragraph. The negative values (convergence) are detected in the part of the SCE, where currents are directed onshore, and positive values (divergence) are detected on the offshore periphery of the eddy. Another area of a strong convergence is observed in the area where coastal waters are entrained in the eddy orbital motion. Such eddies are mostly formed due to interaction of eastward along-shore currents with capes (see more details in). At the same time, cyclonic eddies induce the westward along-shore currents. The confluence of these currents is observed in the area of the current separation from the cape, where we observe the strongest convergence in fig. 4b, 4c. This effect  should promote the entrainment of TSM in the eddy and can explain the maximal value of reflectance on the Landsat images (Figure 4b-left).

Line 212: “stipe” -???

 Response: Excluded from the text

Lines 214-218: does this example support or reject the hypothesis?

Response: This example actually supports the hypothesis because, as in previous cases, the convergence is observed in the coastal part of SCE. We corrected the text to be more clear.

Line 225: change in ‘the orientation’ of the coastline

Response: Thank You. Corrected

Line 226: “Then…” Fig 4 is a static image. It cannot show a process. All the discussion here is pure speculation.

Response: Corrected throughout the text

Line 239: how the eddy “core” is defined?

Response: We agree that this phrase is not accurate and rewrite it

As a result, the inward spiral forms, which cause the propagation of turbid water in the center of such SCE

Fig 5: what are the letters (in black boxes) in the figure?

Response: We extend the description of this figure in the text and add the description of letters.

Case1: Chl is not a passive tracer. The reduction in Chl concentration in the eddy does not necessarily mean mixing with surrounding waters and/or eddy weakening.

Response: We agree with this comment .We excluded this phrase from the text. We also extend a discussion on the possible impact of the biological effects on Chl in SCE

Line 289: “northeast storm”- ? northeast wind?

Response: Replaced on Strong northeast wind

Line 291: “its headland”-???

 Corrected. “On 11 September 2004 the jet separates from the coast rotated cyclonically and formed a first SCE (C1) near the cape Hersones.”

Line: “…is clearly observed…” how? What is the signature of this eddy that we can clearly observe???

Response: We agree that it is not so clear. We extend the description of this eddy. Below You can see the zoomed figures showing this eddy with red arrows highlighting tracers distribution. Such tracers distribution is a usual sign of a vortex dipole in satellite imagery. For the reader, we added red arrows in fig.8a,b to illustrate the manifestation of this larger mesoscale eddy and corrected the text.

“Such trajectories are probably caused by the advection of SCE by a larger mesoscale anticyclonic structure situated to the west of Crimea that is observed in fig. 8a. This mesoscale feature looks like an area with increased TSM in the form of a vortex dipole located to the southwest of to the west of Crimea coast (red arrows in fig. 8a, b).”

Figure R2 Zoomed figure 7a and 7b showing the vortex dipole (red arrows) in the distribution of TSM near the Crimean peninsula

Fig 8b: I do not see any anticyclonic eddy (clearly) in the velocity map.

Response: Due to low resolution of altimetry data, it can not reproduce this small anticyclone , especially near the coast. However, altimetry was able to describe clockwise currents to the west of the Crimean Peninsula, which was caused by the MAE seen in fig. 8a. We insert this comment to the text.

Fig. 9a: why is it relevant? It does not show any comparison with the background currents.

Response: We insert this image to demonstrate the relation of the SCE trajectories on the background currents. This information can be valuable for the reader, as it gives some understanding to the reason of the observed propagation patterns of such SCEs.

Line 313: “ the variability” – the distribution.

Response: We corrected the figure caption.

 Line 338: I do not understand how the “rate of mixing” may depend on the gradient between the eddy and surrounding waters.

Response: We agree that this phrase is purely subjective and exclude it from the text

Lines 381-385: delete – repetition.

Response: Sorry, corrected

Line 396: “bet” – be

Response: Corrected

Line 483: contradicts to line 453 saying that eddies start to decay immediately after formation.

 Response: We agree and rewrite this sentence. This indicates a gradual attenuation of the eddy, which begins immediately after its separation from the coast..

Line 523: “pulsating” – what does that mean?

Response: Replaced on short-period.

The paper would strongly benefit from the careful proofreading. It is not practical to comment on all the details at this point; I will highlight just a few instances:

 ‘in’ satellite images (line 40)

 diffusion ‘of’ momentum (line 114)

“a spots” (line 153). Spots – plural, ‘a’ is not needed.

‘motions (plural)…spin…’ (line 155); common mistake in the text

Response: Thank You. We carefully read the paper and corrected all found grammatical errors and mistakes.

Comments on the Quality of English Language

Careful proofreading is needed. There are multiple gramma errors, including inconsistent use of verb tenses, single/plural, in/on, etc. 

Round 2

Reviewer 1 Report

The authors have addressed most of the comments. 

However, it is still not clear how data assimilation has helped reproduce submesoscale eddies in a high-resolution set-up. Since this studies are heavily based on the assimilated product, I would suggest the authors compare it with the free-run, and investigate if assimilation based on remotely sensed images has made a significant impact on simulating the submesoscale eddies.

None

Author Response

The authors have addressed most of the comments. 

However, it is still not clear how data assimilation has helped reproduce submesoscale eddies in a high-resolution set-up. Since this studies are heavily based on the assimilated product, I would suggest the authors compare it with the free-run, and investigate if assimilation based on remotely sensed images has made a significant impact on simulating the submesoscale eddies.

Response: In fact, we do not use data assimilation in our high-resolution modelling. This is a free-run, which is in agreement with your advice.

The procedure 4d-Var assimilation is a kind of optical flow method, which uses two consecutive satellite images to provide velocity maps. It is not used during high-resolution modelling.

Reviewer 2 Report

The authors have addressed most of the comments. However, two questions remain.
1. As it is said in Discussion section, the SCE _can_ transport suspended sediments. But is there any real possibility to see if this effect is noticeable on the background of general movements of water masses? Are there significant movements of bottom sediments in areas with an increased frequency of eddies?
2. It is sad that the model can not be tested against a real situation, but at least a discussion of the simulation results should be added  clearly marking the similarities with actually observed eddies.

Also, in Figure 12 the dates in the subscription and on the picture are different.

Minor editing of English language required

Author Response

The authors have addressed most of the comments. However, two questions remain.
1. As it is said in Discussion section, the SCE _can_ transport suspended sediments. But is there any real possibility to see if this effect is noticeable on the background of general movements of water masses? Are there significant movements of bottom sediments in areas with an increased frequency of eddies?

Response: Such SCEs may also play an important role in the erosion/abrasion of the coast. Particularly, they can cause resuspension of bottom sediments due to intense vertical shear and, then, redistribute them in the shelf zone. To describe this effect, we need high-frequency data on the vertical distribution of optical properties, which is currently unavailable. We added this information to the text.

On satellite images and in drones data (Kubryakov et al., 2021) we can see that this effect is significant. However, it is relatively short-period, similar to e.g. wind-driven coastal upwelling. It is hard to estimate the role (percentage) of the upwellings in the vertical transport of nutrients, as other factors are important (convective mixing, mesoscale eddies  e.t.c).The same can be said about the role of coastal SCE in the transport of TSM at the moment.

We can mention that another important factor, which causes transport of coastal sediments is along-shore coastal currents. However, they  cause redistribution of TSM near the coast. On contrast, SCE can transport coastal waters offshore and, thereby, can play an important role in the cross-shelf transfer of sediments. The role of the mesoscale eddies is also important in this process and is discussed in the paper.

  1. It is sad that the model can not be tested against a real situation, but at least a discussion of the simulation results should be added  clearly marking the similarities with actually observed eddies.

Response: We agree with this comment and added this information to the discussion.

Also, in Figure 12 the dates in the subscription and on the picture are different.

Response: Thank You. Corrected

Reviewer 3 Report

Thank you for taking my comments into account. The paper is now in a much better shape and can be published. Yet, a few of minor points:  

Figure 1 caption: Please explain how to read probability. Probability of occurrence? I’ve had hard time trying to understand how P=0.12 translates to 35 days per year (line 56).

Line 55: ‘hot points’ - ‘hot spots’ would be better

Lines 254, 544: ‘promontories’ – ‘capes’ would be better (for consistency)

The paper would still benefit from English proofreading.

The paper would still benefit from English proofreading.

Author Response

Thank you for taking my comments into account. The paper is now in a much better shape and can be published. Yet, a few of minor points:  

Figure 1 caption: Please explain how to read probability. Probability of occurrence? I’ve had hard time trying to understand how P=0.12 translates to 35 days per year (line 56).

Response: Sorry, we corrected the numbers in the text: “where P on average is 0.1-0.2, i.e. SCEs are detected 35-70 days per year” In fact, it is probability of the SCE detection, because we may have some uncertainties in the method of eddy identification.

Line 55: ‘hot points’ - ‘hot spots’ would be better

Response: We agree and corrected the text.

Lines 254, 544: ‘promontories’ – ‘capes’ would be better (for consistency)

Response: Thank You. Agree and corrected.